# Characterization and prediction of clinical pathways of vulnerability to psychosis through graph signal processing

Corrado Sandini[1]*, Daniela Zöller[1,2], Maude Schneider[1,3], Anjali Tarun[2], Marco Armando[1], Barnaby Nelson[4,5], Paul G Amminger[4,5,6], Hok Pan Yuen[4,5], Connie Markulev[4,5], Monica R Schäffer[5,6], Nilufar Mossaheb[6], Monika Schlögelhofer[6], Stefan Smesny[6], Ian B Hickie[7], Gregor Emanuel Berger[8], Eric YH Chen[9], Lieuwe de Haan[10], Dorien H Nieman[11], Merete Nordentoft[12], Anita Riecher-Rössler[13], Swapna Verma[14], Andrew Thompson[4,5,15,16], Alison Ruth Yung[4,5,17,18], Patrick D McGorry[4,5], Dimitri Van De Ville[2,19], Stephan Eliez[1,20]

[1]Developmental Imaging and Psychopathology Laboratory, University of Geneva School of Medicine, Geneva, Switzerland; [2]Institute of Bioengineering, Ecole Polytechnique Fédérale de Lausanne, Lausanne, Switzerland; [3]Center for Contextual Psychiatry, Research Group Psychiatry, Department of Neuroscience, KU Leuven, Leuven, Belgium; [4]Orygen, Parkville, Australia; [5]The Centre for Youth Mental Health, The University of Melbourne, Melbourne, Australia; [6]Department of Psychiatry and Psychotherapy, Clinical Division of Social Psychiatry, Medical University Vienna, Vienna, Austria; [7]Department of Psychiatry, University Hospital Jena, Jena, Germany; [8]Brain and Mind Centre, University of Sydney, Sydney, Australia; [9]Child and Adolescent Psychiatric Service of the Canton of Zurich, Zurich, Switzerland; [10]Department of Psychiatry, University of Hong Kong, Hong Kong, China; [11]Department of Psychiatry, Amsterdam University Medical Centers, Amsterdam, Netherlands; [12]Psychiatric Centre Bispebjerg, Copenhagen, Denmark; [13]University of Basel, Basel, Switzerland; [14]Institute of Mental Health, Singapore, Singapore; [15]Division of Mental Health and Wellbeing, Warwick Medical School, University of Warwick, Coventry, United Kingdom; [16]North Warwickshire Early Intervention in Psychosis Service, Conventry and Warwickshire National Health Service Partnership Trust, Coventry, United Kingdom; [17]Division of Psychology and Mental Health, University of Manchester, Manchester, United Kingdom; [18]Greater Manchester Mental Health NHS Foundation Trust, Manchester, United Kingdom; [19]Department of Radiology and Medical Informatics, University of Geneva, Geneva, Switzerland; [20]Department of Genetic Medicine and Development, University of Geneva School of Medicine, Geneva, Switzerland

*For correspondence: corrado.sandini@unige.ch

Competing interests: The authors declare that no competing interests exist.

**Abstract** Causal interactions between specific psychiatric symptoms could contribute to the heterogenous clinical trajectories observed in early psychopathology. Current diagnostic approaches merge clinical manifestations that co-occur across subjects and could significantly hinder our understanding of clinical pathways connecting individual symptoms. Network analysis techniques have emerged as alternative approaches that could help shed light on the complex dynamics of early psychopathology. The present study attempts to address the two main limitations that have in our opinion hindered the application of network approaches in the clinical setting. Firstly, we show that a multi-layer network analysis approach, can move beyond a static

view of psychopathology, by providing an intuitive characterization of the role of specific symptoms in contributing to clinical trajectories over time. Secondly, we show that a Graph-Signal-Processing approach, can exploit knowledge of longitudinal interactions between symptoms, to predict clinical trajectories at the level of the individual. We test our approaches in two independent samples of individuals with genetic and clinical vulnerability for developing psychosis. Novel network approaches can allow to embrace the dynamic complexity of early psychopathology and help pave the way towards a more a personalized approach to clinical care.

## Introduction

Psychiatric disorders are remarkably complex. By the time an individual manifests a sufficient decline in quality of life to warrant consultation with a mental health professional, he will often present a heterogeneous collection of multiple signs and symptoms. The urgent need to provide optimal clinical care, that is evidence-based and consistent across clinicians requires a systematic approach to address such complexity and heterogeneity (*Kendell and Jablensky, 2003*; *McGorry and van Os, 2013*). In particular, clinical practice involves massively reducing dimensionality of information, from the quantitation of up to hundreds of symptoms, to a much more limited number of potential treatment options.

Current approaches to tackle complex clinical patterns in psychiatry have invariably merged together clinical manifestations that tend to co-occur across subjects. The inherent guiding principle is that if two symptoms co-occur in a sufficiently high proportion of patients, their clinical distinction becomes redundant for guiding clinical decision-making. The prototypical example of this approach consists in establishing boundaries within which co-occurrence of psychiatric symptoms is sufficiently high to warrant a single diagnostic label (*Jablensky, 2016*). An alternative dimensional approach consists in progressively merging manifestations of mental health disturbances over progressively higher levels of complexity, on the basis of their empirically observed pattern of co-occurrence (*Caspi et al., 2014*; *Krueger et al., 2018*). The first approach, based on discrete diagnostic categories, is intuitive, and has proven extremely useful in increasing communicability and agreement across clinicians (*Kendell and Jablensky, 2003*). There is, however, growing concern, that merging symptoms into discrete diagnostic labels may be a step too far in reducing the complexity of mental health disturbances (*Maj, 2018*). Moreover, diagnostic algorithms have demonstrated limited utility in guiding therapeutic decisions, which strongly supports the need for reform (*Reed et al., 2018*). More recent dimensional approaches may provide more accurate representation of mental health phenomena. However, their utility in guiding clinical decision-making remains very much debate (*Tyrer, 2018*). Indeed, while 'dimension fit the data,' it is still unclear whether 'clinicians can fit dimensions' (*Tyrer, 2018*).

The implicit assumption that justifies merging clinical manifestations into diagnostic labels or dimensions is that symptoms manifest together as consequence of a common underlying disease mechanism. It has been argued that such underlying conceptualization could be at the origin of the growing dissatisfaction toward dimensional and categorical approaches (*Borsboom, 2017*; *Borsboom and Cramer, 2013*). Indeed, it is increasingly recognized that, in psychiatry, symptoms are not only passive expression of common underlying disease processes. Psychiatric symptoms can often represent active agents, that have the ability to provoke their reciprocal emergence, through dynamic causal interactions (*Borsboom, 2017*; *Borsboom and Cramer, 2013*). For instance, the observation that in patients with chronic psychosis, thought disorders tend to co-occur with social retreat could be explained by the fact that early sub-clinical paranoid ideation hindered the subsequent maintenance of functional social interactions. This is consistent with the concept of secondary negative symptoms (*Carpenter et al., 1985*). Similarly, a causal association between early insomnia and subsequent mood disturbances could partially account for their co-occurrence in depressed patients (*Franzen and Buysse, 2008*).

Importantly, pathways of interactions between individual symptoms are probably not constrained within current diagnostic boundaries, particularly in the earliest stages of psychopathology (*McGorry and van Os, 2013*; *McGorry et al., 2018*; *McGorry et al., 2017*; *van Os, 2013*). For instance, in at risk-populations, sub-threshold psychotic symptoms increase the likelihood of developing not only a full-blown psychotic disorder, but also mood, anxiety, and substance use disorders

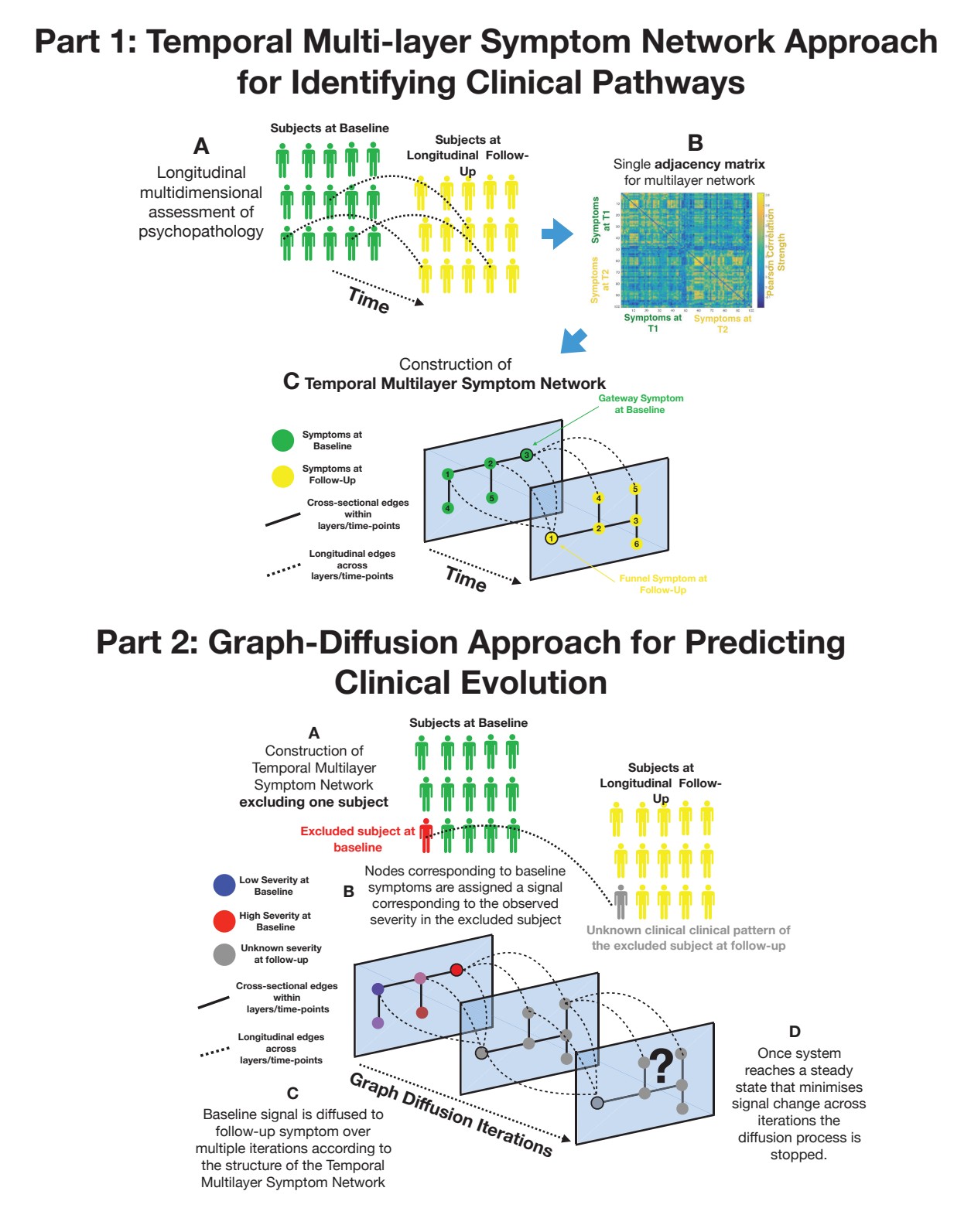

**Figure 1.** Part 1: Methodological pipeline for construction of Temporal MultiLayer Symptom Network (TMSN). (**A**) Clinical assessment of multiple symptoms is performed for a cohort of participants over two time points (baseline and follow-up). (**B**) A single adjacency matrix is constructed, containing cross-sectional correlations between symptoms measured at baseline and follow-up which are located respectively in the upper left and lower right quadrants. The off-diagonal quadrant is composed of longitudinal correlations between each symptom at baseline and each symptom at

*Figure 1 continued on next page*

Figure 1 continued

follow-up. (**C**) Graphical representation of the multilayer symptom network. A first network layer is composed of correlation between baseline symptoms, represented in green and a second layer is composed of correlations between symptoms at follow-up, represented in yellow. The two cross-sectional layers are connected by longitudinal edges composed of correlations between baseline and follow-up symptoms represented as dashed lines. Such intuitive graphical representation is achieved empirically through topological embedding of symptoms according to dimensions derived from Eigen-decomposition of the multilayer adjacency matrix. Graph theory is employed to identify longitudinal clinical pathways (shortest paths) connecting symptoms across temporal layers. Baseline symptoms that broadly influence symptoms at follow-up, have high longitudinal centrality, and can be conceptualized as gateways of psychopathology (schematically represented as symptom three at baseline). Follow-up symptoms with high longitudinal centrality are broadly influenced by symptoms at baseline, and can be conceptualized by funnels of psychopathology (schematically represented as symptom one at follow-up). Part 2: Methodological pipeline for Graph Diffusion-based prediction of clinical evolution. (**A**) A TMSN is reconstructed excluding the individual for whom clinical prediction is performed, in a leave-one-out cross-validation loop. (**B**) In a graph-signal-processing framework node corresponding to baseline symptoms are assigned a signal that corresponds to their empirically observed severity in the excluded individual. At the beginning of the diffusion process severity of symptoms at follow-up is considered to be unknown and corresponding nodes are assigned a value of 0. (**C**) Using a finite-difference graph diffusion approach signal corresponding to the observed clinical pattern at baseline is diffused on the TMSN. Compared to simple regression analysis prediction keeps into account both the structure of longitudinal correlations connecting layers of baseline and follow-up symptoms, which are represented schematically as dashed lines, and the structure of cross-sectional correlations between symptoms at follow-up. This diffusion approach leads to a progressive evolution of the predicted symptom pattern at follow-up over multiple diffusion iterations. The symptom pattern at baseline is considered to be known, and is hence re-initialized at each diffusion iteration. This is conceptually similar to modeling the spread of information in a social network as a function of friendship ties between individuals or the diffusion of temperature as a function of distance between spatial locations. (**D**) For temperature, information or psychopathology, the diffusion algorithm will evolve the predicted signal until the system converges toward an equilibrium that minimizes signal change across time, at which point prediction is considered to be stable the iterative diffusion will be stopped. This process was repeated to predict symptom severity at follow-up for each individual included in the cohort, in a leave-one-our cross-validation loop.

(*Lin et al., 2015*; *Rutigliano et al., 2016*). On the opposite, the presence of affective and amotivation symptoms strongly increase the likelihood of conversion to psychosis in individuals with psychotic symptoms (*McGorry et al., 2017*; *Dominguez et al., 2010*). In the field of developmental psychopathology, cross-disorder interactions are particularly prominent, where they have been described as sequential comorbidity (*Caron and Rutter, 1991*; *Taurines et al., 2010*). Such cross-diagnostic clinical trajectories currently represent a major challenge for decision-making in early intervention psychiatry (*McGorry et al., 2018*).

Clinical evidence presented above points to two considerations, which will be relevant for designing novel approaches to the assessment and classification of patients. First, given that individual symptoms often play an active role in determining clinical trajectories, merging symptoms together into diagnostic labels or disease dimensions, could significantly hinder our understating of disease mechanisms at stake. This, in turn, could limit our ability to tailor treatment protocols to the individual patients need (*McGorry et al., 2018*; *van Os, 2013*). Second, given that clinical trajectories are heterogenous, novel approaches to the assessment and classification of patients, should be broad and trans-diagnostic, particularly in the early stages of psychopathology (*McGorry and Nelson, 2016*). Still, modeling pathways of interactions between up to hundreds of symptoms represent a significant computational challenge. Novel analysis techniques will be required in order to translate such complex multidimensional information to the clinical setting, in a way that can reliably inform decision-making. The importance of developing novel data-analysis approaches, specifically conceived for clinical application, will only increase with the advent of digital 'precision-medicine' approaches to characterize behavior in psychiatry (*Torous and Baker, 2016*; *Insel, 2018*). The hope is that the tools of data science will allow to embrace the complexity of mental health problems, and ultimately assist in a personalized approach to clinical care (*Torous and Baker, 2016*).

Network science is a rapidly expanding branch of mathematics dedicated to the study of graphs, which can be broadly defined as structures composed of discrete nodes that are connected by edges (*Newman, 2010*). Applications of network science range from the study of networks of social interactions (*Wasserman and Faust, 1994*) to that of networks of biological interaction between genetic transcripts (*Chen et al., 2014*). From a computational perspective, the core appeal of network analyses is the ability to represent broad patterns in the overall structure of a data set, while conserving highly granular information relative to individual variables. For instance, in terms of the overall structure, network techniques can identify the propensity groups of variables that are more densely connected to each other than to rest of the network, which is defined as network

modularity. Moreover, network techniques offer several indices to measure connectivity profiles of individual variables. For instance, the propensity for an individual variable to strongly mediate the relationship between other network variables can be defined as a network centrality.

The application of network analysis carries the potential to have a profound theoretical and practical impact on the study of mental health disturbances. Indeed, according to the network theory of psychopathology, mental disorders are best conceptualized, as systems of reciprocally interacting symptoms (*Rubinov and Sporns, 2010*; *Boccaletti et al., 2006*). The most widely implemented paradigm has consisted of measuring correlations between different pairs of psychiatric symptoms in cross-sectional samples, reconstructing a network of symptoms-symptoms interactions (*Borsboom and Cramer, 2013*; *Rubinov and Sporns, 2010*; *Boccaletti et al., 2006*). Within a network perspective, clinical manifestations that have a high tendency to co-occur across subjects, and that might otherwise be merged together in single dimension or diagnostic label, would instead be represented as a highly interconnected network module, composed of individual symptoms. By conserving the singularity of individual clinical manifestations, it is then possible to identify symptoms that play a particularly prominent role in mediating the relationship between other clinical variables, and that are said to have high network centrality. High centrality is commonly considered to reflect a prominent causal role in influencing other symptoms.

Network approaches are rapidly gaining popularity by demonstrating that exploiting the high-dimensional granularity of clinical assessments can generate insights that would be missed if symptoms were merged in diagnoses or dimensions (*Robinaugh et al., 2020*). For instance, the application of network analysis techniques to the study of schizotypal personality revealed that subclinical forms of paranoia and of behavioral or thought disorganization play a particularly central role in influencing the presence of other schizotypal personality traits (*Christensen et al., 2018*; *Fonseca-Pedrero et al., 2018*). Moreover, social anhedonia played a central role in mediating the relationship between positive and negative schizotypal personality traits (*Christensen et al., 2018*). In health-seeking individuals, network analysis revealed that prodromal symptoms mediated the relationship between basic symptoms and frank symptoms of psychosis, with disorganized communication again playing a particularly central role (*Jimeno et al., 2020*). Finally, by using network analysis, studies have shown that clinical manifestations that are not necessarily specific to psychosis, including in particular affective dysregulation, may play a prominent role in influencing the presence of psychotic-like experiences, in adolescence (*Fonseca-Pedrero et al., 2021*).

Identifying pathways of interactions between individual symptoms carries significant potential, in terms of assisting in predicting prognosis, and planning treatment strategies. Still, despite

**Table 1.** Correspondence of items of SIPS and CAARMS clinical interviews.

| Corresponding items of SIPS | Corresponding items of CAARMS | Missing items of SIPS | Missing items of CAARMS |
|---|---|---|---|
| P1 Unusual thought | 1.1 Unusual thought content | D2 Bizarre thinking | 3.3 Inadequate affect |
| P2 Persecutory ideas | 1.2 Non-bizarre ideas | D4 Personal hygiene | 4.1 Alogia |
| P4 Pperceptual abnormalities | 1.3 Perceptual abnormalities | G1 Sleep disturbances | 5.1 Social isolation' |
| P5 Disorganized communication | 1.4 Disorganized speech | | 5.4 Aggressive behavior |
| N1 Social anhedonia | 4.3 Anhedonia | | 6.2 Objective motor functioning |
| N2 Avolition | 4.2 Avolition/apathy | | 6.3 Subjective bodily sensation |
| N3 Expression emotion | 3.2 Blunted affect | | 6.4 Subjective autonomic functioning |
| N4 Experience emotion | 3.1 Subjective emotional disturbance | | 7.1 Mania |
| N5 Ideational richness | 2.2 Objective cognitive change | | 7.3 Suicidality/self-harm |
| N6 Occupational functioning | 5.2 Impaired role functioning | | 7.4 Affective instability |
| D1 Odd behavior | 5.3 Disorganized behavior | | 7.5 Anxiety |
| G2 Dysphoric mood | 7.2 Depression | | 7.6 OCD |
| G3 Motor disturbances | 6.1 Subjective motor functioning | | 7.7 Dissociative symptoms' |
| G4 Impaired tolerance to stress | 7.8 Impaired subjective tolerance to normal stress | | BPRS grandiosity |
| D3 Trouble attention | 2.1 Subjective cognitive change | | |

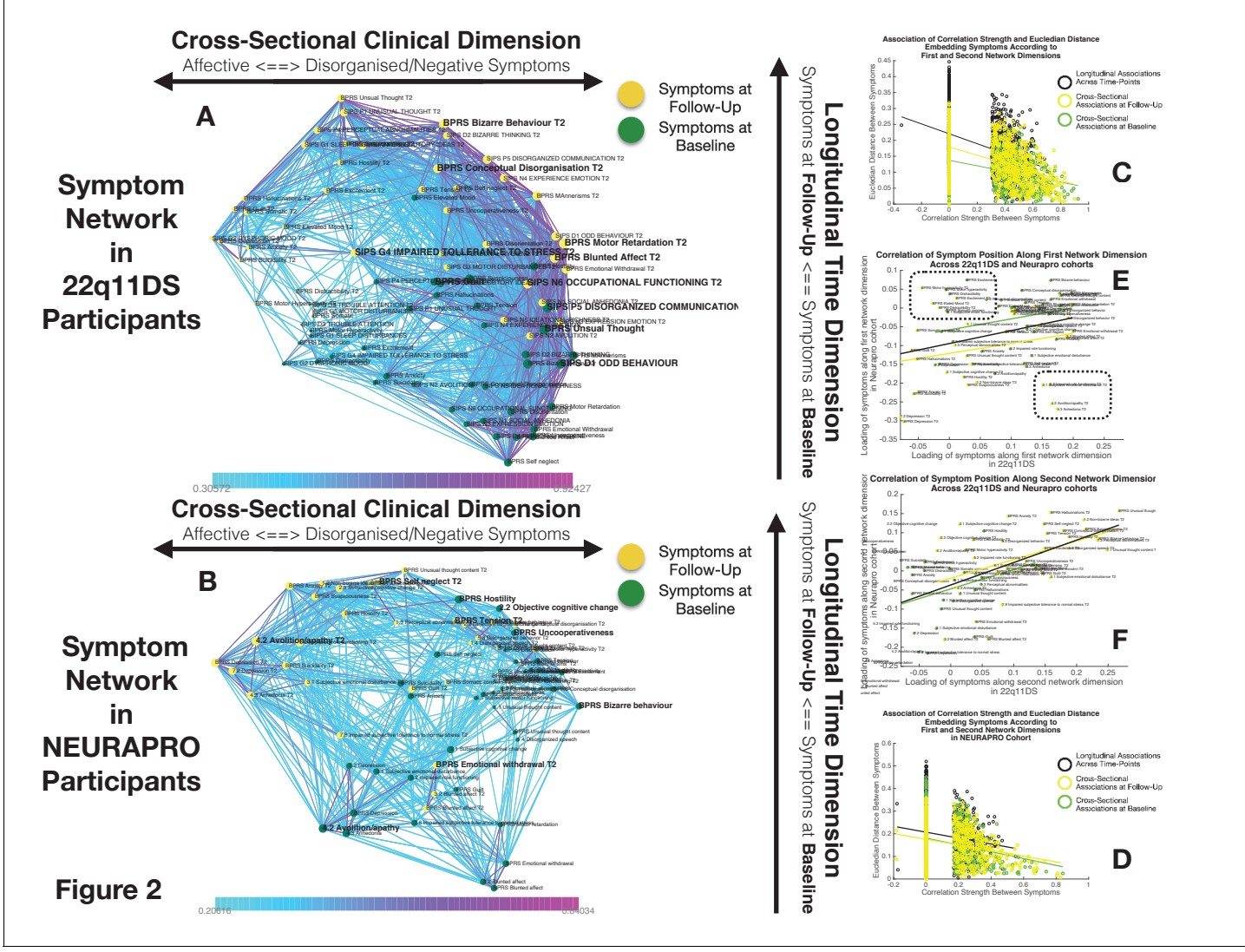

**Figure 2.** Structure of Longitudinal Symptom Networks. (A, B) Structure of longitudinal symptom networks in 22q11DS (A) and NEURAPRO samples (B). Spatial embedding of symptoms according to network dimensions derived from singular value decomposition. The first dimension is plotted along the horizontal X-axis whereas the second dimension is plotted along the vertical Y-axis. Lines connecting symptoms represent correlations that survive correction for multiple comparisons at p<0.05 color-coded according to the correlation strength. Symptoms at baseline are displayed in green and symptoms at follow-up are displayed in yellow. Size of nodes is scaled according to the mean connectivity strength of each symptom. Symptoms that present a higher than random centrality in mediating clinical pathways going from baseline to follow-up are displayed in bold. (C, D) Association of Euclidean distance between symptoms after spatial embedding according to the first and second network dimensions and empirically observed correlation strength between symptoms in 22q11DS (C) and NEURAPRO samples (D). Cross-sectional associations between symptoms at baseline are displayed in green and between symptoms at follow-up are displayed in yellow. Longitudinal associations between symptoms at baseline and symptoms at follow-up are displayed in black. (E) Association between the position of symptoms according to the first network dimension across 22q11DS and NEURAPRO cohorts. Two clusters of symptoms that contribute negatively to the correlation between structures of symptom networks across the two cohorts, suggesting a different pattern of correlation with other forms of psychopathology, are circled. (F) Association between the position of symptoms according to the second network dimension across 22q11DS and NEURAPRO cohorts.

considerable promise, network approaches have to date largely remained confined to the laboratory. Below, we suggest that recent methodological advances made in the fields of dynamic network analysis and graph-signal-processing can help to address three main obstacles to the clinical translation of network approaches to psychopathology.

The first shortcoming is that computational challenges have largely limited the application of network techniques to cross-sectional data. As a consequence, psychiatric symptoms networks typically lack the essential dimension of time. For instance, high centrality in a cross-sectional sample could

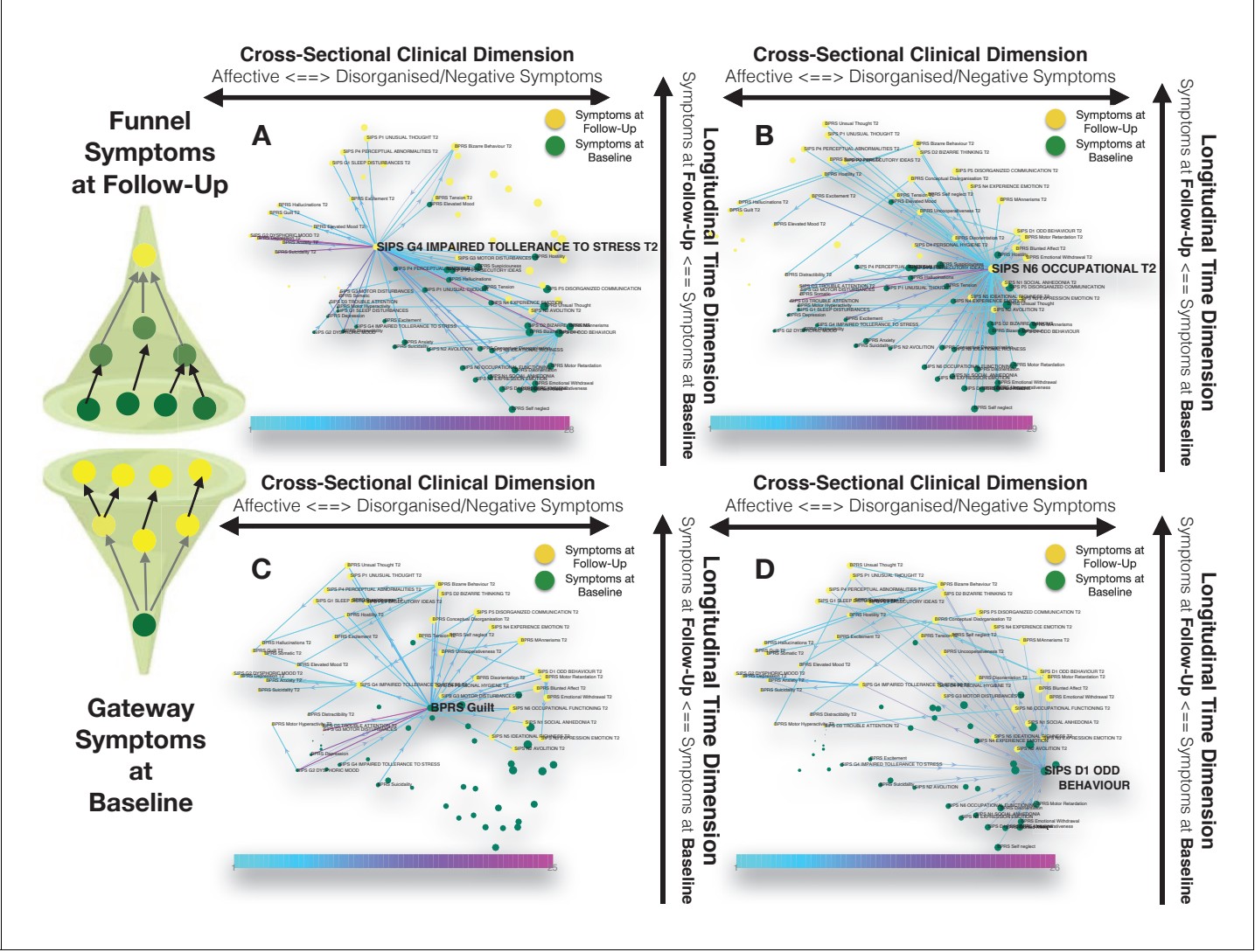

**Figure 3.** Longitudinal clinical pathways running through gateway symptoms at baseline (C, D) and funnel symptoms at follow-up (A, B) in the 22q11DS sample. (A) Impaired tolerance to daily stress at follow-up acts as a funnel by broadly mediating the effects of baseline of thought disturbances on follow-up affective symptoms and of baseline affective symptoms on follow-up thought disturbances. (B) Reduced occupational functioning at follow-up, acted as a funnel by broadly mediating the effects of negative, disorganized symptoms and ADHD symptoms at baseline on the persistence of negative and disorganized symptoms at follow-up. (C) BPRS guilt at baseline acts as a gateway by mediating the effects of affective symptoms at baseline on both affective and thought disturbance symptoms at follow-up. (D) SIPS Odd Behavior acted as a gateway by broadly mediating the effects of negative symptoms at baseline on both disorganized and negative symptoms at follow-up.

imply that a symptom has an active role in broadly influencing subsequent clinical manifestations. However, an opposite but equally likely interpretation is that high centrality reflects the tendency of symptoms to be passively influenced by different prior psychiatric manifestations. To address this limitation, we propose a temporal multilayer symptom network (TMSN) approach mutated from dynamic network analysis (*Mucha et al., 2010*). A TMSN, applied to developmental psychopathology, would consist of a first temporal layer composed of cross-sectional correlations between symptoms at a first baseline assessment. The subsequent network layers are composed of correlations between symptoms measured at longitudinal follow-ups. Such cross-sectional layers would be connected by longitudinal edges reflecting the association of symptoms across time, namely which symptoms at baseline predicted which symptoms at follow-up. Analytic tools of network science could then allow the dissection of longitudinal disease pathways connecting manifestations of psychopathology over time (*Mucha et al., 2010*). For instance, it would be possible to dissect

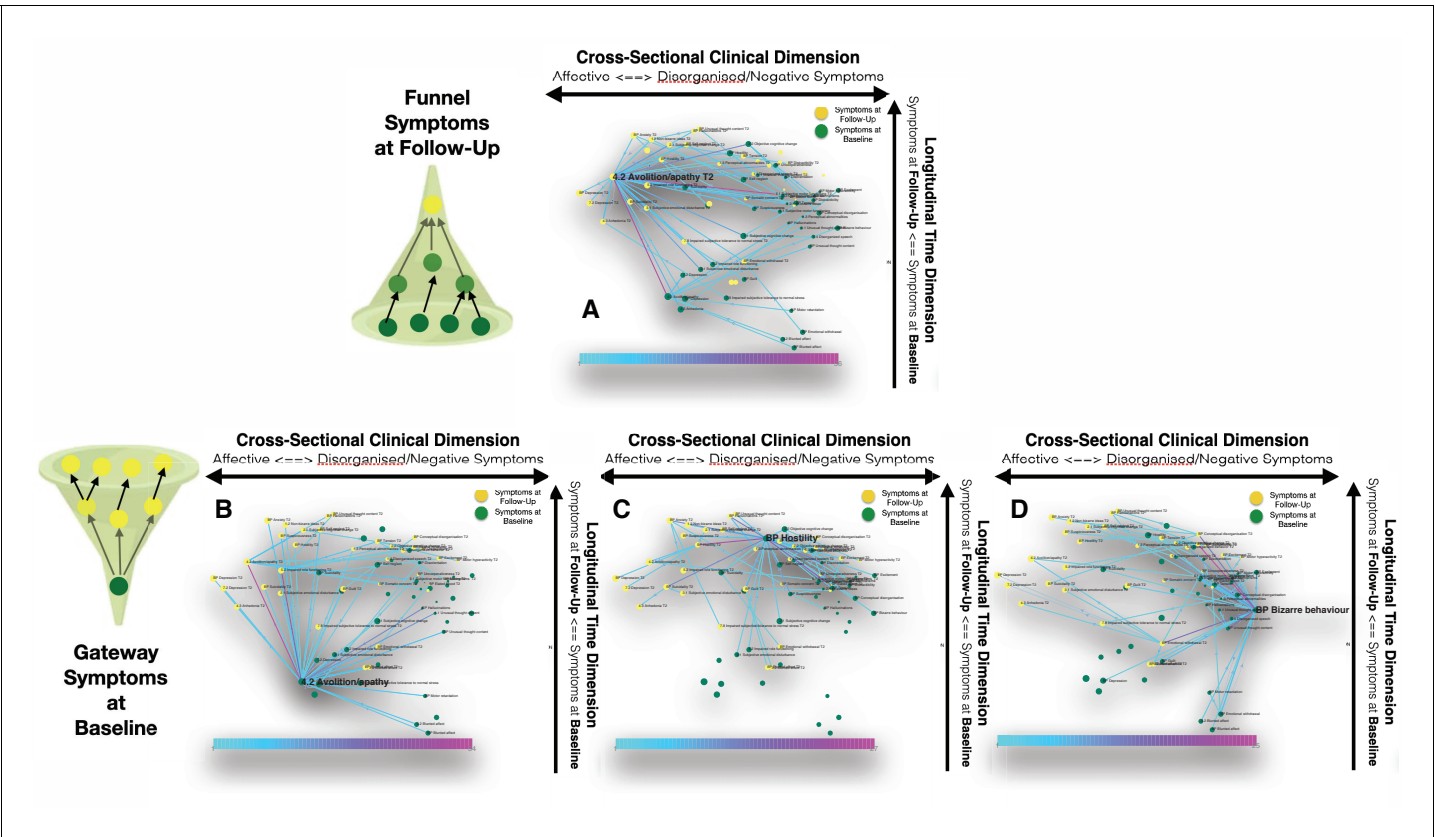

**Figure 4.** Longitudinal clinical pathways running through gateway symptoms at baseline (**B**, **D**) and funnel symptoms at follow-up (**A**) in the NEURAPRO sample. (**A**) Avolition-apathy at follow-up was also highlighted as a key funnel symptom that broadly mediated the effects of baseline avolition and affective disturbances on subsequent psychopathology. (**B**) Avolition-apathy was located on the left side of the graph and was directly associated with subsequent affective symptoms and indirectly associated with negative and disorganized symptoms, through the mediating role of persistent avolition-apathy at follow-up. (**C**) BPRS hostility was also located in proximity to negative and disorganized symptoms at baseline and appeared central in mediating their effects on subsequent symptoms of mood disturbance. (**D**) BPRS bizarre behavior was located on the right side of the graph and appeared to broadly affect negative and disorganized symptoms at follow-up. Moreover, bizarre behavior indirectly affected subsequent affective disturbances p through the mediating role of emotional withdrawal at follow-up.

symptoms at baseline that broadly affect clinical manifestations at follow-up, and can be conceptualized as *gateways* of psychopathology, from symptoms at follow-up that are broadly affected by psychopathology at baseline, acting as *funnels* of psychopathology (see *Figure 1*).

A second limitation is that current representations of network structure arguably remain excessively complex. In the near future, the pursuit of 'high-definition' personalized medicine in psychiatry is likely to provide an even greater wealth of information regarding factors that influence the dynamic evolution and interaction between symptoms (*Torkamani et al., 2017*). For instance, experience-sampling techniques and digital phenotyping will allow the monitoring of fluctuations of multiple clinical, environmental or physiological variables in a daily life setting (*Myin-Germeys et al., 2018*). Network analysis techniques are ideally suited, and are indeed being implemented, to analyze such rich information, which carries tremendous clinical potential (*Myin-Germeys et al., 2018*). Crucially, however, for this complex high-dimensional information to translate into clinical practice, results will not only need to be statistically significant, but should also be intuitively accessible and interpretable. Indeed intuitiveness and communicability remain the main advantages of current diagnostic systems (*Tyrer, 2018*). An ideal framework would need to balance a quantitative low-resolution characterization of the structure of psychopathology similar to factorial analysis, with high-resolution information regarding relevant pathways of interaction between individual symptoms. From this perspective, an approach that seems particularly promising is the use of techniques of dimensionality reduction on graphs, to achieve a topological embedding of individual symptoms

that reflects the most salient aspects of the overall network architecture (*Alanis-Lobato et al., 2016*). Employing such topological embedding to a multilayer temporal network of symptoms could offer an intuitive characterization of clinical pathways contributing to the evolution of psychopathology.

The third, and arguably major obstacle to clinical translation, is that psychopathology networks characterize symptom connectivity at the population level, whereas in clinical practice decisions are made on the basis of symptom severity, at the level of the individual. Despite considerable promise, no study to date has, to the best of our knowledge, demonstrated the utility of network approaches in predicting the dynamic development of psychopathology and assist in establishing prognosis. Graph signal processing (GSP) is a relatively novel field of network science that is interested in moving beyond the quantitative characterization of network architecture to model how network architecture affects processes that occur on the network (*Shuman et al., 2013*). Similarly, to other branches of network science, GSP is devoted to analyzing graphs composed of nodes connected by edges, such as graphs composed of individuals connected by social ties. The unique aspect of GSP is that each node in the graph can be assigned a signal such as the amount of information in a social network, or symptom severity in a psychopathology network. Techniques of GSP can then allow us to study and predict how diffusion of signal on the graph (i.e., diffusion of information among individuals) is influenced by the architecture of connections between nodes (i.e., architecture of social bonds) (*Shuman et al., 2013*). With regard to psychopathology, techniques of GSP seem extremely attractive to model how dynamic interactions between multiple symptoms will influence heterogeneous clinical evolutions. Specifically, once interactions between symptoms are modeled as a multilayer temporal network, GSP could allow the prediction of how network architecture will influence the diffusion of psychopathology across temporal layers at the level of individual patients.

In the present study, we implemented tools of multilayer network analysis and GSP to characterize and predict clinical pathways of vulnerability to psychopathology in two longitudinal samples of individuals characterized as being at high risk of developing a psychotic disorder. The first sample is composed of individuals with 22q11.2 Deletion Syndrome (22q11DS), a homogenous genetic disorder, associated with an approximately 30% risk of developing a psychotic disorder (*McDonald-McGinn et al., 2015*; *International Consortium on Brain and Behavior in 22q11.2 Deletion Syndrome et al., 2014*). The second sample was composed of individuals at clinical high risk for developing psychosis, recruited from 10 centers internationally in the context of a clinical trial to test the efficacy of polyunsaturated fatty acids (PUFAs) (*Nelson et al., 2018*). This first objective was to attempt to provide a quantitative and at the same time intuitive representation of clinical pathways of interaction between symptoms contributing to clinical evolution. The second objective was to use network interactions between symptoms to predict clinical evolution at the level of individual participants. For each section we begin by presenting results in the 22q11DS cohort, followed by results of the replication analysis performed in clinical high-risk individuals.

## Materials and methods

### Sample and clinical instruments

#### Primary cohort of individuals with 22q11DS

Individuals with 22q11DS were part of a prospective longitudinal study that has been described in several previous publications (*Sandini et al., 2018*; *Schaer et al., 2009*). Recruitment was performed through patient associations and word of mouth in French and English-speaking European countries. Inclusion criteria for the overall longitudinal study were the presence of a genetically confirmed 22q11.2 deletion and willingness of the participant and caregiver to participate in the study. Exclusion criteria for the overall longitudinal study were the inability of the participant to follow the procedures described in the project due to sensory issues (e.g., blindness) or too severe cognitive impairments. In particular, participants should have sufficient verbal skills to comprehend and answer to oral questions (i.e., during clinical interviews). For the present study, specific inclusion criteria were the availability of two longitudinal clinical assessments, including a first baseline assessment during adolescence, defined between 11 and 19 years of age. Presence of a psychotic disorder at baseline according to DSM-IV-TR criteria was an exclusion criterion. This yielded a total of 57 individuals (M/F=26/31), for whom a first psychiatric assessment was available during adolescence (age

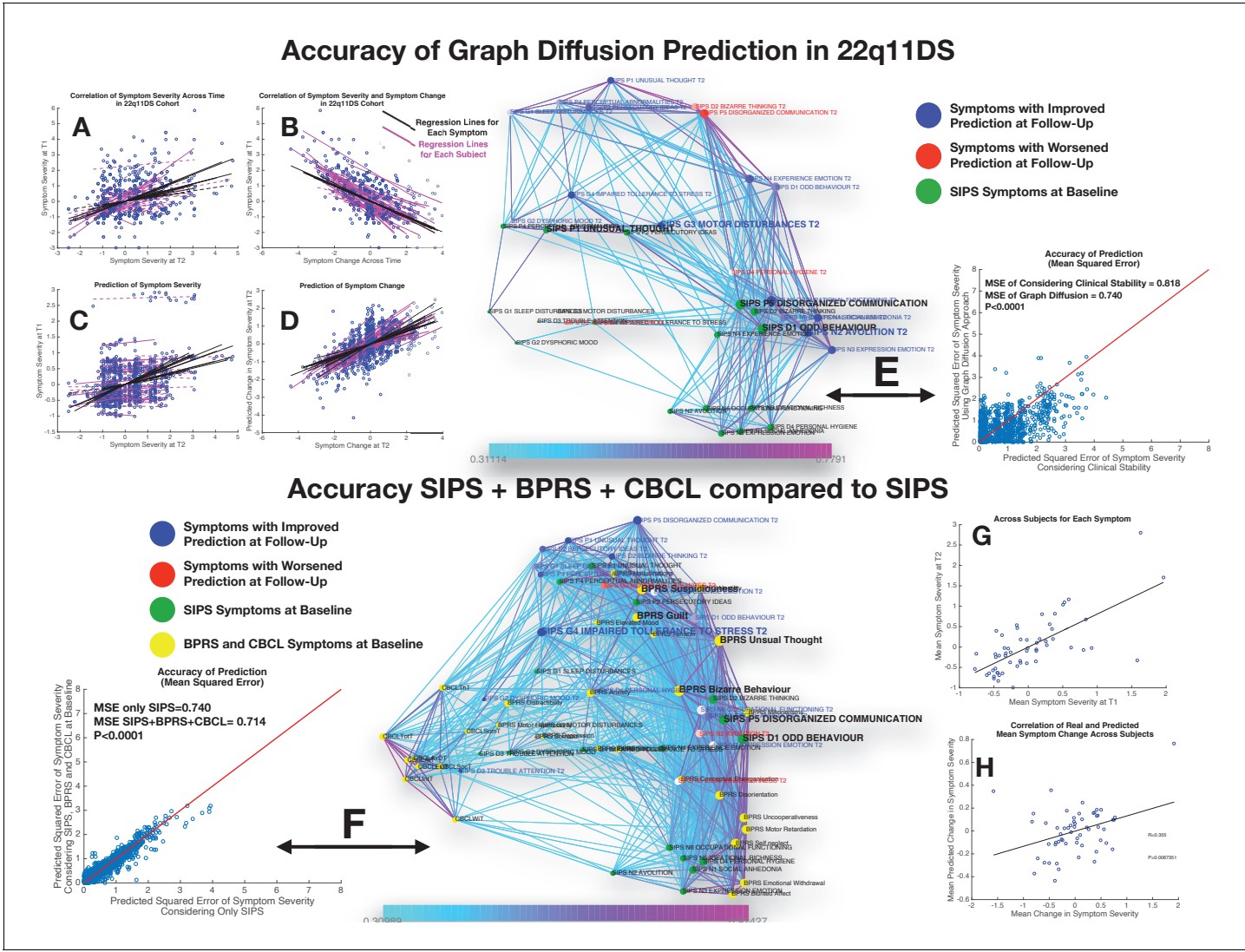

**Figure 5.** Performance of graph diffusion approach in predicting patterns of SIPS psychopathology at longitudinal follow-up in the 22q11DS sample. (A) Correlation of symptom severity across time points for all symptoms across participants. Regression lines for symptoms are displayed in black with dashed lines indicating correlations that are not significant at p<0.05. Regression lines for individuals are displayed in purple with dashed lines indicating correlations that are not significant at p<0.05. (B) Correlation between symptom severity at baseline and change in symptom severity between baseline and follow-up for all symptoms across all participants. (C) Correlation between real and predicted symptom severity at follow-up. (D) Correlation between real and predicted symptom change between baseline and follow-up. (E) Comparison of accuracy in predicting SIPS at follow-up, between considering clinical stability and graph diffusion approach using SIPS at baseline. Symptoms are spatially embedded according to two main network dimensions derived from SVD. Symptoms at baseline are displayed in green. Symptoms at follow-up are color-coded according to the prediction accuracy of graph diffusion compared to considering clinical stability, with blue symptoms having higher accuracy using graph diffusion and red symptoms having worsened accuracy. (F) Prediction accuracy of considering the combination of SIPS, BPRS, and CBCL at baseline compared to using only items of the SIPS. Symptoms are spatially embedded according to two main network dimensions derived from SVD. Symptoms of the SIPS at baseline are displayed in green. Items of additional clinical instruments are displayed in yellow. Symptoms at follow-up are color-coded according to the prediction accuracy of considering an additional clinical instrument compared to the accuracy achieved by using only items of the SIPS, with blue symptoms having higher accuracy and red symptoms having worsened accuracy. (G) Correlation between mean symptom severity at follow-up and mean predicted symptom severity at follow-up. (H) Correlation between mean symptom change across time points and mean predicted symptom change.

range at baseline 11.6–18.4, mean 14.4±1.8) along with a second longitudinal assessment on average 3.8±1 years later (age range at follow-up 14.2–24.27, mean 18.25±2.0).

Psychiatric diagnoses were assessed with the Diagnostic Interview for Children and Adolescents-Revised and the psychosis supplement from the Kiddie Schedule for Affective Disorders and

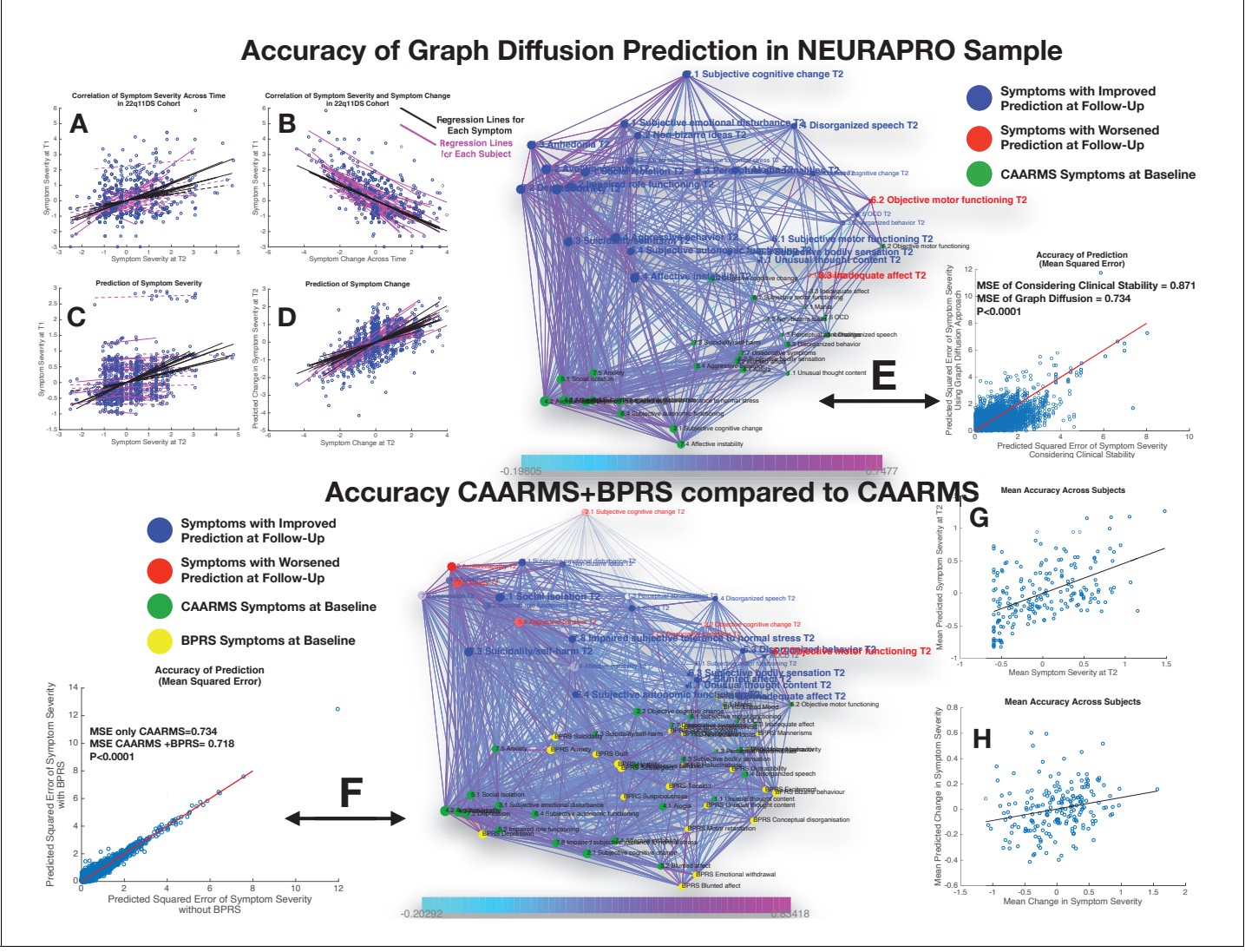

**Figure 6.** Performance of graph diffusion approach in predicting patterns of CAARMS psychopathology at longitudinal follow-up in the NEURAPRO sample. (A) Correlation of symptom severity across time points for all symptoms across participants. Regression lines for symptoms are displayed in black with dashed lines indicating correlations that are not significant at p<0.05. Regression lines for individuals are displayed in purple with dashed lines indicating correlations that are not significant at p<0.05. (B) Correlation between symptom severity at baseline and change in symptom severity between baseline and follow-up for all symptoms across all participants. (C) Correlation between real and predicted symptom severity at follow-up. (D) Correlation between real and predicted symptom change between baseline and follow-up. (E) Comparison of accuracy in predicting CAARMS at follow-up, between considering clinical stability and graph diffusion approach using CAARMS at baseline. Symptoms are spatially embedded according to two main network dimensions derived from SVD. Symptoms at baseline are displayed in green. Symptoms at follow-up are color-coded according to the prediction accuracy of graph diffusion compared to considering clinical stability, with blue symptoms having higher accuracy using graph diffusion and red symptoms having worsened accuracy. (F) Prediction accuracy of considering the combination of CAARMS and BPRS at baseline compared to using only items of the CAARMS. Symptoms are spatially embedded according to two main network dimensions derived from SVD. Symptoms of the CAARMS at baseline are displayed in green. Items of BPRS at baseline are displayed in yellow. Symptoms of CAARMS at follow-up are color-coded according to the prediction accuracy of considering an additional clinical instrument compared to the accuracy achieved by using only items of the CAARMS, with blue symptoms having higher accuracy and red symptoms having worsened accuracy. (G) Correlation between mean symptom severity at follow-up and mean predicted symptom severity at follow-up. (H) Correlation between mean symptom change across time points and mean predicted symptom change.

Schizophrenia Present and Lifetime Version for individuals below 18 years of age (*Kaufman et al., 1997*; *Reich, 2000*). For adult participants, we used the Structured Clinical Interview for DSM-IV Axis I Disorders (*First MB et al., 1996*).

To assess sub-threshold positive, negative, disorganized, and generalized psychotic symptoms, individuals completed the Structured Interview for Prodromal Syndromes (SIPS) (*Miller et al., 2002*). For a broad characterization of psychopathology, we employed the Brief Psychiatric Rating Scale (BPRS) (*Overall and Gorham, 1962*). To quantify global measures of severity of psychopathology, we employed a combination of the parent-reported versions of the Child Behavior Checklist (CBCL) and Adult Behavior Checklist (ABCL) (*Achenbach, 2003*; *Tm, 1991*). Full clinical characterization was performed at both baseline and longitudinal follow-up.

For the primary construction of multilayer symptom networks, we initially considered items of the SIPS and BPRS instruments measured at baseline and longitudinal follow-up. We removed symptoms that had a non-zero score in less than 1% of the sample leading to the exclusion of SIPS grandiosity and BPRS grandiosity scales. This yielded a total of 41 clinical measures available at both baseline and follow-up. All clinical variables were available for all subjects included in the study and we did not exclude any outliers.

## Replication in individuals at Clinical Ultra High Risk for Psychosis in NEURAPRO cohort

The second cohort of individuals, without a confirmed 22q11.2 Deletion, but meeting criteria for Clinical Ultra High Risk for Psychosis, was recruited in the context of the NEURAPRO clinical trial, designed to test effects of $\omega-3$ PUFA therapy (*McGorry et al., 2017*; *Nelson et al., 2018*). Individuals were recruited among help-seeking populations in Australia, Singapore, Italy, Germany, Hong Kong, Denmark, and Switzerland. Inclusion criteria have been described in detail in previous publications and yielded a total of 304 subjects with a clinical UHR status at baseline. Once included in the study, individuals were randomized to a double-blind 6-month treatment with either $\omega-3$ PUFA or placebo, and were then followed up for further 6 months, yielding a total follow-up period of 12 months (*McGorry et al., 2017*; *Nelson et al., 2018*). Specific inclusion and exclusion criteria are detailed in previous publications (*Cruz et al., 2017*) and yielded an overall sample of 304 individuals.

From the original sample of 304 individuals, we excluded 18 subjects with the missing assessment of at least one item of the Comprehensive Assessment of At-Risk Mental Sate (CAARMS) at baseline, one of which subsequently converted to psychosis. Another 89 subjects were excluded due to missing full characterization at the 12-month follow-up (79 missing CAARMS items, 6 missing BPRS), 19 of which converted to psychosis. This yielded a total of 201 individuals (M/F=98/103) with full clinical characterization at both baseline and 12-month follow-up (age range at baseline: 13.3–37.8 mean 20±4.5). Excluded subjects were not significantly different from the rest of the sample in terms of severity of any item the available CAARMS assessment at baseline or follow-up. However, a higher proportion of individuals who converted to psychosis lacked a full clinical assessment at longitudinal follow-up, as revealed by a higher proportion of individuals who converted to psychosis among excluded subjects compared to subjects included in our analysis (19 conversions to psychosis/103 excluded subjects by 17 psychosis conversions/201 included subjects, P-value Chi-Square test=0.004; see *Supplementary files 1* and *2* for details ).

Psychiatric diagnoses were determined with the Structured Clinical Interview for DSM-IV-TR Axis I Disorders (*First MB et al., 1996*). Sub-threshold positive, negative, and generalized psychotic symptoms were evaluated with the Comprehensive Assessment of the At-Risk Mental State (*Yung et al., 2005*). The BPRS was employed for a broad characterization of psychopathology (*Overall and Gorham, 1962*) and the Montgomery-Asberg Depression Rating Scale (MADRS) was employed to measure depressive symptoms (*Montgomery and Asberg, 1979*).

Directly comparing network structure across 22q11DS and NEURAPRO cohorts was complicated by the use of different SIPS and CAARMS semi-structured clinical interviews across the two samples. Both interviews are designed to assess clinical high risk for developing psychosis, with similar operationalized diagnostic criteria and comparable predictive value (*Fusar-Poli et al., 2016*). Still, there is no one-to-one correspondence between each item of the two scales. We hence referred to the two manuals to define items that had sufficiently high correspondence across the two instruments. Based on this assessment, we excluded three symptoms that were considered as specific on the SIPS in the 22q11DS and 13 symptoms that were considered as specific on the CAARMS in the NEURAPRO sample. This yielded a total of 37 shared items across the two populations considering both SIPS/

CAARMS and BPRS instruments. These items were used to construct longitudinal symptom networks (see *Table 1*).

## Statistical analysis pipeline

### Multilayer symptom networks to define clinical pathways of vulnerability

#### Construction of multilayer symptom networks

Prior to constructing networks, we accounted for the effects of age and sex with linear regression. We then constructed a single multilayer symptom network for each sample, in which each node represented a symptom, and the connecting edge between symptoms was weighted by the Pearson correlation between the two corresponding symptoms across subjects. Graph edges (i.e., correlations between symptoms) were initially computed cross-sectionally at both baseline and follow-up, composing two separate temporal layers. Such separate temporal layers were connected by longitudinal edges estimated from the correlations between symptoms at baseline and symptoms at follow-up, producing a single multilayer temporal network. Such multilayer network can be expressed in a single adjacency matrix composed of both cross-sectional and longitudinal correlations (see *Figure 1*).

We thresholded the network by considering only correlations survived correction for multiple comparisons with false discovery rate (FDR) at p<0.05 using Benjamini-Yekutieli procedure as implemented in Matlab (*Benjamini and Yekutieli, 2005*). As a supplementary analysis, we also constructed networks employing a range of more stringent connectivity thresholds both in 22q11DS and NEURAPRO samples. Results of such analysis are reported in *Appendix 1—figures 4* and *5* and indicate an overall stability of network structure over a range of connectivity thresholds.

Networks were constructed considering both significant positive and negative correlations. However, in order to facilitate interpretability of differences in connectivity strength among different network edges, networks represented in the main text include only significant positive correlations. Significant negative correlations (one correlation in 22q11DS sample and eight correlations in NEURAPRO sample) are represented separately in *Appendix 1—figures 6* and *7*.

## Network topological embedding

Arguably one of the main challenges of current network models relates to intuitiveness and interpretability of results. Our objective was hence to provide a low-dimensional, easily interpretable, visual representation of the multilayer network that still reflected main patterns of correlations between symptoms, both within and across time.

To derive such main correlation patterns, we extracted main dimensions of variance in the network, by conducting eigendecomposition on the thresholded adjacency matrix, representing the multilayer symptom network. Then, network nodes were spatially embedded according to their loading along the two first principal network components.

This procedure yielded a two-dimensional spatial representation that groups together symptoms that are closely connected in the multilayer network. Indeed, eigenvectors of the network provide a *low-dimensional* representation of the main correlation structure between symptoms. However, by simply using these low-dimensional components for the spatial embedding and keeping every symptom as a single node, we retain the *high-dimensional* characterization of the relationships between specific clinical symptoms, both within each time point and across longitudinal time points.

The procedure employed to choose the appropriate number of network components is described in detail in the supplementary material. It revealed that, for both samples, only the first three principal components explained higher proportion of variance that what would be expected in a network with random structure. Spatial embedding of symptoms according to the third network dimension however did not offer a meaningful representation of the relationship between symptoms across time and is reported in *Appendix 1—figures 9* and *10* for 22q11DS and Neuropro samples, respectively.

To test whether spatial embedding of symptoms according to the first two principal network dimensions provided a meaningful representation of network structure we correlated the strength of correlations between symptoms with their Euclidean distance in two-dimensional space. We expected to observe an overall negative correlation between Euclidean distance and correlation strength, indicating that symptoms there were strongly correlated, both with and across time points,

tended to cluster together in space. We verified that negative association between Euclidean distance and correlation strength, was present for both correlations and longitudinal correlation, which would indicate that spatial embedding reflected the structure of both the cross-sectional and longitudinal relationship between symptoms. Moreover, we correlated topological embedding of symptoms according to the two main network dimensions across samples, in order to have an estimate of the degree of similarity of network structure across samples.

## Graph theory analysis of longitudinal clinical pathways

Spatial embedding of symptoms provided an intuitive representation of the major patterns of relationships between symptoms. We were then interested in complementing this representation with a quantitative characterization of longitudinal pathways of interactions between symptoms across time.

To do this, we employed the tools of graph theory, which is a branch of mathematics that is specifically devoted to the study of graphs, and that has provided multiple quantitative tools to quantify the connectivity profiles of both the overall network and of individual symptoms/nodes. We specifically employed the graph-theoretical tools implemented in the Brain-Connectivity-Toolbox for Matlab (The MathWorks, Inc, Natick, MA; http://www.brain-connectivity-toolbox.net). First, the size of the symptoms in the network was scaled according to their overall connectivity strength. This means that larger symptoms had an overall stronger cross-sectional and longitudinal correlation with the rest of the symptoms in the network.

As a supplementary analysis, we employed the predictability algorithm (*Haslbeck and Fried, 2017*), using the procedure described in *Haslbeck and Waldorp, 2018*, to estimate how well each symptom could be predicted by the rest of the symptoms in the multilayer network. Predictability was estimated for both symptoms and baseline and symptoms at follow-up considering both cross-sectional and longitudinal relationships between symptoms. Results are reported in *Appendix 1—figure 1*.

We were then interested in focusing on pathways of longitudinal interaction between symptoms across time. To do this, we employed the Floyd-Warshall algorithm, implemented in the brain-connectivity toolbox for Matlab, to identify the shortest clinical paths connecting each symptom at baseline with each symptom at longitudinal follow-up. We then derived a longitudinal betweenness centrality measure by counting the number of longitudinal clinical paths running through each individual symptom. Such longitudinal centrality measure can be conceptualized as the relative importance of each clinical variable in mediating the relationship between symptoms at baseline and at follow-up, across time. In order to identify symptoms with higher longitudinal betweenness centrality than expected by chance, we constructed 10,000 random networks matched for connectivity by reshuffling edge position. We computed shortest paths connecting symptoms across time in each random network deriving a null distribution of longitudinal betweenness centrality. P-values for each symptom were computed by estimating the probability of observing a higher betweenness centrality measure than this empirical null distribution. Further, we used the FDR at p<0.05 to correct for multiple comparisons.

First, this approach identified symptoms at baseline that over-proportionately mediated effects on symptoms at follow-up. Such longitudinal network hubs at baseline can be conceptualized as *gateways* of psychopathology. Second, our approach identified symptoms at follow-up that were over-proportionately affected and mediated the effects of symptoms at baseline. Such longitudinal network hubs at follow-up can be conceptualized as *funnels* of psychopathology.

## Graph diffusion approach to predict patterns of clinical evolution

Current network approaches to psychopathology have focused on studying the architecture of interactions between psychiatric symptoms mostly by employing techniques of graph theory (*Borsboom, 2017*; *Borsboom and Cramer, 2013*). It should however be noted that, in a graph theory framework, symptoms are characterized purely in terms of their connectivity profile with other nodes/symptoms. For network approaches to inform clinical practice at the level of individual patients, symptoms would need to be characterized not only in terms of how they interact with each other, but also in terms of their severity. Indeed, an ideal framework would exploit knowledge of

network interactions between symptoms to help predict the evolution of symptom severity across time.

GSP is different from graph theory in that, aside from studying the architecture of network connections, nodes can be assigned a value or signal (*Shuman et al., 2013*). Once nodes are assigned a signal in a GSP framework, graph diffusion algorithms have been developed to model how graph architecture influences the propagation of such signals across nodes (*Shuman et al., 2013*). An intuitive implementation of this approach is to predict how variations in temperature diffuse over time, across multiple discrete spatial locations. The dynamics of temperature propagation will be determined by the reciprocal distance between spatial locations, with positions that are closer in space having a higher likelihood to influence their neighbor's temperature, over short periods of time. Graph diffusion addresses this computational problem in a network construct, by modeling discrete spatial locations as nodes in distances as the inverse of connectivity strength between multiple nodes of a network. This then allows predicting how topological network structure influences the dynamics of temperature diffusion.

Graph diffusion approaches are increasingly demonstrating their potential in medical applications. For instance, applying graph diffusion to a multilayer network has been shown to predict the relationship between genetic mutations and tumor samples (*Timilsina et al., 2019*). Moreover, studies are hinting at the potentials of this approach to model disease progression. Indeed, Raj et al. showed that modeling the spread of dementia-related neuropathological alterations as a function of the network architecture of long-range axonal fiber bundles reliably predicts the empirically observed patterns of brain atrophy (*Raj et al., 2012*). To the best of our knowledge, graph diffusion approaches have however not yet been employed in the study of psychopathology. Here, we propose that applying graph diffusion approaches to multilayer temporal symptoms network can predict the evolution of symptom severity across time, at the level of individual patients.

Our methodological approach, described schematically in *Figure 1*, began by constructing a multilayer temporal symptom network, excluding the subject for whom we attempted to predict clinical evolution, in a leave-on-out cross-validation loop. As previously described the multilayer temporal symptom network was composed both of cross-sectional correlations between symptoms with each time point and longitudinal correlations between baseline and follow-up symptoms. At the beginning of the diffusion process, symptoms at baseline were assigned a signal that corresponded to the severity that was empirically observed in the excluded subject. Severity of symptoms at follow-up was considered to be unknown and their signal was initially set to 0. The signal corresponding to the empirically observed clinical pattern at baseline was then diffused on the multilayer temporal symptom network, in order to predict symptom severity at follow-up. This is conceptually similar to predicting temperature propagation according to the network structure of distance between nodes.

To predict the spread of symptom severity from baseline to follow-up symptom we employed an iterative finite-difference graph diffusion approach. Compared to simple regression analysis, this approach considers both longitudinal correlations across time points and cross-sectional correlations between symptoms at follow-up, leading to a progressive evolution and refinement in the predicted symptom pattern. In the example of heat diffusion, the temperature distribution at Time 1 is considered fixed (and therefore re-imposed at each iteration of the algorithm), while the distribution at Time 2 evolves by the diffusion process. For both temperature and psychopathology, the diffusion algorithm will evolve the predicted signal until the system converges toward an equilibrium that minimizes signal change across time, at which point the iterative diffusion will be stopped. The graph diffusion converges to a steady-state solution upon reaching a minimal signal change between iterations that is less than 1e−9. Once such threshold was achieved, the clinical prediction for symptom severity at follow-up was considered to be stable, and the diffusion process was stopped. This process was repeated to predict symptom severity at follow-up for each subject included in the cohort, in a leave-one-out cross-validation loop.

In mathematical terms, the diffusion equation follows a linear differential equation given as follows:

$$\frac{\partial x(t)}{\partial t} = -\gamma L x(t)$$

If we approximate the solution to this differential equation through numerical methods, a finite-difference approach is employed, wherein in this case, the Laplacian matrix acts as a difference

operator. One can therefore iteratively apply the Laplacian operator to the signal at each time point t until we reach a stable solution x(*tf inal*). Algebraically, this approach can be implemented by solving the differential equation directly in order to arrive at a closed-form solution in terms of matrix exponential given as follows:

$$x = \exp(-\gamma L t) x_0$$

This solution is a negative exponential term, which decays for large values of t. Therefore, the solution is guaranteed to reach a stable form, and hence the convergence of the algorithm is guaranteed.

The procedure employed to evaluate the algorithm's accuracy in predicting clinical evolution is described in the results section.

## Results

### Structure of multilayer symptom networks and longitudinal clinical pathways of vulnerability in 22q11DS

Topological embedding of symptoms yielded a strong negative correlation between the Euclidean distance separating symptoms and the empirically observed correlation strength (R=−0.465, p<0.0001), observed not only for cross-sectional associations between symptoms at baseline (R=−0.354, p<0.0001) or at follow-up (R=−0.473, p<0.0001), but also for longitudinal associations between symptoms at baseline and at follow-up (R=−0.365, p<0.0001; see *Figure 2C*). As shown in *Appendix 1—figure 10A*, such associations between Euclidean distance and correlation strength remained significant even after restricting data-range by excluding negative correlations. These results suggest that an easily interpretable low-dimensional embedding can offer a good approximation of the structure of the multilayer symptoms network.

The first network dimension, plotted along the horizontal axis in *Figure 2A*, mainly captured the structure of cross-sectional correlations between symptoms within each time point. Symptoms located on the right side of the graph mainly captured disorganization and thought disorder including SIPS Odd Behavior and Disorganized communication and BPRS Bizarre Behavior, Mannerism, and Unusual Thought Content. Negative symptoms were mostly located on the right side of the graph, near disorganization symptoms. The opposite left side of the graph was, on the other hand, populated by symptoms of affective dysregulation, including SIPS Dysphonic Mood and Reduced Tolerance to Normal Stress and BPRS Depression and Anxiety. Symptoms of attention deficit hyperactivity disorder (ADHD) including SIPS Trouble with Attention, BPRS Distractibility, and Motor Hyperactivity, were located on the left side of the graph near affective disturbances. Positive symptoms had an intermediate position along the first dimensions, with SIPS Perceptual Abnormalities and BPRS Hallucinations being closer to affective and ADHD symptoms, whereas SIPS and BPRS Thought Disorder were closer to negative and disorganized symptoms. Loading of symptoms along this first eigenvector was highly correlated across time points (R=0.7, p<0.0001), pointing an overall stability in the cross-sectional structure of the symptom network over time, along with an affective to negative/disorganized dimension.

The second dimension was plotted along the vertical Y-axis and predominantly captured the temporal aspect, with symptoms at baseline located at the bottom of the graph and symptoms at follow-up being located at the top of the graph. Importantly, aside from an overall distinction of symptoms across time points, we observed a significant variation along the time dimension between symptoms measured within each time point, which captured the differential propensity of symptoms to influence one another over time. Indeed, we observed an opposite association across the two time points between loading of symptoms according to the second time dimension and the mean strength of longitudinal correlations between symptoms at baseline and at follow-up (at baseline R=0.3, p=0.05, at follow-up R=−0.22, p=0.16, p of difference=0.0094; see *Appendix 1—figure 1C*). In this perspective, symptoms that were higher than average at baseline can be considered more highly predictive of psychopathology at follow-up. On the opposite, symptoms that were located lower than the rest at follow-up, were more directly influenced by prior psychopathology at baseline. This representation offered an intuitive characterization of the relationship between symptoms over time.

Subsequently, we were interested in highlighting clinical pathways involving individual symptoms that played a particularly prominent role in disease progression. Our approach based on graph theory identified four symptoms at baseline that disproportionately affected clinical symptom patterns at follow-up, and that can be conceptualized as gateways of psychopathology (shown in bold in *Figure 2A*). Network embedding presented before provided an intuitive characterization of the different longitudinal clinical pathways affecting such gateway symptoms. The first three symptoms were located on the right side of the graph and mainly captured thought disorder and disorganization including SIPS Disorganized Communication, SIPS Odd Behaviour, and BPRS unusual thought. Disorganization symptoms, such as SIPS Odd Behavior, acted as a gateway by broadly mediating the effects of negative symptoms at baseline on both disorganized and negative symptoms at follow-up (see *Figure 3D*). A fourth gateway symptom was represented by BPRS guilt. BPRS guilt was located closer to the left side of the graphs and acted as a gateway by broadly mediated the effects of affective symptoms at baseline on both affective and thought disturbance symptoms at follow-up (see *Figure 3C*).

Our approach also identified six symptoms at follow-up, that were broadly affected by psychopathology at baseline, and that can hence be conceptualized as funnels of psychopathology. Two of these funnel symptoms were captured disorganization and were represented by bizarre behavior and conceptual disorganization, and mostly mediated the effects of prior disorganization symptoms. Two more were represented by negative symptoms such as BPRS blunted effect and SIPS occupational functioning, which were located on the right side of the graph and appeared to importantly mediated the effects of negative and disorganized symptoms and ADHD symptoms at baseline on the persistence of negative and disorganized symptoms at follow-up (see *Figure 3B*). A final funnel was represented by SIPS reduced tolerance to normal stress, which located left side of the graph appeared important in mediating the effects of baseline of thought disturbances on follow-up affective symptoms and of baseline affective symptoms on follow-up thought disturbances (see *Figure 3A*).

As a confirmatory analysis, we also constructed multilayer temporal networks, employing Spearman's rank correlations, which are displayed in *Appendix 1—figure 3*. Results pointed to a strong similarity of networks reconstructed from Pearson versus Spearman correlations, both in terms of the loading of symptoms across the two main network dimensions and in terms of longitudinal centrality of individual symptoms. Still, while most symptoms were identified as longitudinal hubs in both networks, conceptual disorganization at follow-up was identified as a significant longitudinal Hub, after correction for multiple comparisons, only in the Pearson correlation network. Baseline symptoms of suspiciousness and impaired tolerance and follow-up symptoms of somatic disturbance and mannerisms were significant only in Spearman networks.

## Structure of multilayer symptom networks and longitudinal clinical pathways of vulnerability in NEURAPRO sample

While the variance explained by the first two dimensions was lower in the NEURAPRO sample, we still observed a significant negative correlation between the Euclidean distance separating symptoms and the empirically observed correlation strength (R=−0.249, p<0.000; see *Figure 2D*), observed for both cross-sectional associations between symptoms at baseline (R=−0.249, p<0.0001) or at follow-up (R=−0.238, p<0.0001), and for longitudinal associations between symptoms at baseline and at follow-up (R=−0.135, p<0.0001). As shown in *Appendix 1—figure 10B*, such associations between Euclidean distance and correlation strength remained significant even after restricting data range by excluding negative correlations. This suggests that spatial embedding of symptoms according to the two main eigenvectors still offered a meaningful characterization of the interaction between individual symptoms.

Similar to results in 22q11DS, the first dimension mainly captured variance between cross-sectional correlations within each time point (see *Figure 2B*). Symptoms located on the right side of the graph were mostly composed of positive and disorganized symptoms, including Bizarre Behavior, Unusual Thought, and Hallucinations measured with both BRPS and CAARMS. The left side of the graph was mostly populated by symptoms of affective disturbances, including Depression and Anxiety, BPRS Guilt, and CAARMS Subjective Reduced Tolerance to Daily Stressors. Negative symptoms could be divided into two subgroups according to their loading along the first dimension. Specifically, symptoms of reduced emotional expressiveness, such as Blunted Affect, BPRS Emotional

withdraw, and CAARMS Anhedonia, were located on the right side of the graph, closer to disorganized symptoms. On the opposite symptoms of reduced motivational drive, such CAARMS Avolition/Apathy and Impaired Role Functioning, were located closer to the left side of the graph and closer to anxiety/depressive symptoms. Interestingly, we observed a significant positive correlation between the loading of symptoms along the first 'cross-sectional' dimension (R=0.299, p=0.008) across 22q11DS and NEURAPRO samples (see *Figure 2E*). This would suggest a similar structure of cross-sectional psychopathology across 22q11DS and NEURAPRO samples, mainly reflecting an overall distinction of affective and negative-disorganized symptoms. Still, in the context of an overall similar network structure, two groups of symptoms appeared to cluster differently in the networks of the two samples (circles in *Figure 2E*). In particular, symptoms of ADHD, including BPRS motor hyperactivity and distractibility, were in proximity to affective symptoms in 22q11DS, whereas they were closer to symptoms of thought disorder in the NEURAPRO sample. Moreover, a sub-group of negative symptoms, including experience of emotion, avolition, social anhedonia, and occupational functioning, was located closer to other negative and disorganized symptoms in the 22q11DS cohort whereas they clustered closer to depressive and affective symptoms in the NEURAPRO sample.

The second network dimension mainly captured the dimension of time, with symptoms at baseline mainly located at the bottom of the graph and symptoms at follow-up mainly located at the top of the graph. Similarly, what was found in the 22q11DS cohort, there was significant variance within symptoms at each time point along this time dimension. Interestingly, the correlation of symptom loading across samples was even stronger along this second longitudinal dimension (R=0.56, p<0.0001; see *Figure 2F*), suggesting that the relative predictive value of symptoms at baseline in influencing symptoms at follow-up, and the relative tendency of symptoms at follow-up to be influenced by prior psychopathology, is similar across the two clinical populations.

Our approach identified three baseline symptoms that presented disproportionately high centrality in mediating clinical patterns at follow-up displayed in *Figure 4*. In particular, BPRS bizarre behavior was located on the right side of the graph and appeared to broadly affect negative and disorganized symptoms at follow-up. Moreover, bizarre behavior indirectly affected subsequent affective disturbances p through the mediating role of emotional withdrawal at follow-up. BPRS hostility was also located in proximity to negative and disorganized symptoms at baseline and appeared central in mediating their effects on subsequent symptoms of mood disturbance. Finally, avolition-apathy was located on the left side of the graph and was directly associated with subsequent affective symptoms and indirectly associated with negative and disorganized symptoms, through the mediating role of persistent avolition-apathy at follow-up. Indeed avolition-apathy at follow-up was also highlighted as a key funnel symptom that broadly mediated the effects of baseline avolition and affective disturbances on subsequent psychopathology.

As a confirmatory analysis, we constructed multilayer temporal networks in the NEURAPRO sample, employing Spearman's rank correlations, which are displayed in *Appendix 1—figure 2*. Similar to what observed in 22q11DS, results pointed to a strong similarity of networks reconstructed from Pearson versus Spearman correlations, both in terms of the loading of symptoms across the two main network dimensions and in terms of longitudinal centrality of individual symptoms. Minor discrepancies across Spearman versus Pearson correlation networks included the fact that BPRS symptoms of Hostility at baseline and Emotional Withdrawal at follow-up were identified as hubs only in the Pearson network whereas centrality of BPRS Uncooperativeness was significant only in the Spearman network, after FDR correction for multiple comparisons.

Despite an overall similar network structure appeared similar in 22q11DS and NEURAPRO cohorts, we did not observe a significant association in measures of longitudinal betweenness centrality (R=−0.03, p=0.77). These results suggest that specificities exist in the role of individual symptoms in contributing to the evolution of psychopathology, across the two samples.

## Graph diffusion approach to predict patterns of clinical evolution

### Evaluation of prediction accuracy in 22q11DS and NEURAPRO cohorts

Our primary objective was to predict the multivariate patterns of symptoms included in the SIPS and CAARMS clinical interviews, designed to assess vulnerability to psychosis. We started by predicting the severity of SIPS and CAARMS items at follow-up using items of SIPS and CAARMS at baseline.

Subsequently, we estimated the added benefit of considering additional clinical instruments at baseline.

Simply correlating severity at baseline against severity at follow-up for each symptom across individuals, revealed a positive significant correlation both in the 22q11DS cohort for 14/18 symptoms being tested (R=0.37±0.12) and in the NEURAPRO cohort for 27/28 symptoms (R=0.31±0.1; see *Figures 5A* and *6A*, respectively). While not surprising, these results suggest that simply considering a patient as clinically stable across time points provides a highly non-random estimate of the clinical pattern at follow-up. We computed the mean squared change in symptom severity across time points across all individuals and symptoms. This mean-squared-error is a measure of prediction accuracy achieved by simply considering clinical stability across time points, that we used as a baseline against which we tested the performance of our graph diffusion-based prediction approach.

A perhaps less intuitive observation was that change in symptom severity between baseline and follow-up was strongly negatively correlated with symptom severity at baseline for all symptoms being tested in both the 22q11DS (R=−0.65±0.17) and NEURAPRO cohorts (R=−0.63±0.25), suggesting the existence of a phenomenon of regression to the mean (see *Figures 5B* and *6B*, respectively).

## Performance of prediction in 22q11DS sample

Considering only the SIPS subscale at baseline yielded a significant prediction of SIPS symptom severity at follow-up, as revealed by a strongly significant correlation between actual and predicted symptom severity (R=0.40, p<0.00001) across all items and individuals, that remained significant when averaging mean and predicted symptom severity in each subject (R=0.64, p<0.00001; see *Figures 5C* and *4E*). Interestingly, the correlation between empirical and predictive values was even stronger when considering symptom change across the two time points for all symptoms and individuals (R=0.57, p<0.00001). However, when averaging symptom change in each subject, we did not observe a significant correlation between observed and predicted values (R=−0,22, p=0.08). In other terms, the algorithm predicted both mean and specific symptoms severity at follow-up and specific change in symptom severity, while it failed to predict the mean change in symptom severity (see *Figure 5F*).

Importantly, prediction accuracy of graph diffusion was significantly higher than simply considering clinical stability (MSE of clinical stability=0.818±0.82, MSE of graph diffusion=0.74±0.62, p<0.00001; see *Figure 3G*). Exploring the distribution in the difference of prediction accuracy across symptoms revealed that accuracy of graph diffusion was higher for all symptoms except personal hygiene, bizarre thinking, disorganized communication, and trouble with attention (see *Figure 5H*).

Next, we were interested in assessing the added value of considering additional clinical instruments at baseline. Adding the BPRS evaluation at baseline provided a small but significant improvement in SIPS prediction at follow-up (MSE of SIPS=0.74±0.62, MSE of SIPS+BPRS=0.72±0.60, p=0.03), whereas adding the CBCL at baseline did not significantly improve the accuracy of symptom prediction at follow-up (MSE of SIPS=0.74±0.62, MSE of SIPS+CBCL=0.73±0.60, p=0.316). However, when considering the combination of adding CBCL+BPRS, this yielded a strong increase in prediction accuracy that was highly significant compared to considering only the SIPS (MSE of SIPS=0.74±0.62, MSE of SIPS+BPRS+CBCL=0.716±0.59, p<0.00001) or separately adding BPRS (p<0.00001), or CBCL (p<0.00001) (see *Figure 3C*). Moreover, in addition to significantly predict mean symptom severity (R=0.69, p<0.00001), relative symptom severity (R=0.45, p<0.00001), and relative symptom change (R=0.61, p<0.00001), adding BPRS and CBCL significantly predicted mean change in SIPS symptom severity over time (R=0.35, p=0.006; see *Figure 5C and D*).

These results point to a synergism of BPRS and CBCL in predicting clinical patterns of the SIPS at follow-up. Interestingly, this synergism was visually apparent from the position of the items of the two instruments within the structure of the longitudinal symptom graph. Indeed, while items of the CBCL clustered on the left side of the graph in proximity to affective and ADHD symptoms, most items of the BPRS were located on the right side of the graph in proximity to thought disorder and negative symptoms.

## Performance of prediction in NEURAPRO sample

Similarly, to what was observed in the 22q11DS, the graph diffusion approach yielded a significant prediction of patterns of symptom severity at follow-up, with an average correlation between real and predicted symptom severity across all individuals (R=0.26, p<0.0001; see *Figure 6C*). Correlation was stronger between real and predicted symptom change between baseline and follow-up (R=0.54, p<0.0001; see *Figure 6D*). When averaging severity across symptoms for each individual we observed a significant correlation between mean and predicted symptoms severity (R=0.51, p<0.0001) but not between mean and predicted change in symptom severity (R=0.04, p=0.56), similar to what was observed in the 22q11DS cohort (see *Figure 6E and F*). We hence compared the prediction accuracy of the graph diffusion approach against that of simply considering clinical stability across time. As in 22q11DS, this analysis revealed that MSE of the graph diffusion approach was significantly lower than simply considering clinical stability (MSE of clinical stability=0.871±0.79, MSE of graph diffusion=0.734±0.67, p<0.00001; see *Figure 6G*). Indeed, accuracy of prediction was higher for all items of the CAARMS except inadequate effect, objective motor functioning, and mannerism (see *Figure 6H*).

Next, we estimated the additive predictive value of considering additional clinical instruments at baseline. Adding the BPRS at baseline provided a strong improvement in the prediction of CAARMS items at follow-up (MSE of CAARMS=0.734±0.67, MSE of CAARMS+BPRS=0.718±0.66, p<0.0001; see Figure 7C). Moreover in addition to significantly predict mean symptom severity (R=0.50, p<0.0001), relative symptom severity (R=0.25, p<0.00001), and relative symptom change (R=0.57, p<0.00001), adding BPRS significantly predicted the mean change in CAARMS symptom severity over time (R=0.29, p=0.001; see *Figure 6* ). On the other hand, adding MADRS scores did not significantly improve average prediction accuracy (MSE of CAARMS=0.734±0.67, MSE of CAARMS+MADRS=0.734±0.67, p=0.589). Moreover, considering the addition of BRPS+MADRS worsened the accuracy of prediction compared to the combination of CAARMS and BPRS (MSE of CAARMS+BPRS=0.718±0.66, MSE of CAARMS+MADRS=0.723±0.67, p<0.0001). The lack of added predictive value of the MADRS to the CAARMS would have been predicted from the position of the MADRS items at baseline within the structure of the longitudinal symptom network. Indeed, although MADRS items were located on the left 'affective' side of the graph, they were located lower along the time dimension that corresponds to affective items of the CAARMS. This would suggest that the CAARMS characterization of affective dysregulation at baseline is sufficient and indeed superior to MADRS items in terms of predicting CAARMS psychopathology at follow-up.

## Discussion

Current clinical approaches to tackle the complexity of mental health disturbances have almost invariably merged together clinical manifestations that often co-occur across participants. However, especially in the earliest stages of psychopathology, merging clinical manifestations may hinder our understanding of pathways of interaction between individual symptoms, which in turn may be relevant for predicting prognosis or planning treatment strategies. Network approaches to psychopathology represent a promising framework to model complex disease pathways between individual symptoms, but two main factors may have to date limited their clinical translation.

The first limitation refers to the insufficient intuitiveness and interpretability of results of current network analyses. We argue that such insufficient interpretability is the combined result of the application of network approaches to cross-sectional data, together with the excessive complexity of resulting symptom networks. In the present study, we propose a methodological approach based on multilayer network analysis that offers an intuitive and quantitative and quantitative characterization of clinical pathways of interaction between symptoms over time.

The second main limitation is that current network approaches characterize symptoms exclusively in terms of their reciprocal interactions, which are estimated at the level of a population. Clinical practice on the other hand entails making predictions about symptom severity at the level of the individual. Here, we propose that a network approach inspired by GSP can allow to combine information regarding symptom *connectivity* and *severity* allowing to predict multivariate patterns of clinical evolution at the level of individual participants.

We test our approach in two independent samples of individuals at risk for developing psychosis.

# Temporal multilayer symptom network approach to characterize clinical pathways of vulnerability to psychopathology

A prerequisite for interpreting the role of specific symptoms is having a broad characterization of the overall structure of psychopathology, similar to seeing the outline of the forest before focusing on the trees. In both samples, the first network component captured to the overall cross-sectional structure of relationships between symptoms, largely reflected a distinction between affective versus negative-disorganized psychopathology. Such cross-sectional structure was conserved both across longitudinal visits and across samples and is consistent with results of classical factorial analysis in both high-risk populations and schizophrenia (*McGrath et al., 2004*; *Stefanovics et al., 2014*; *Lançon et al., 1998*; *Wallwork et al., 2012*). This would suggest that overall network architecture reflects broad clinical patterns observed in clinical practice, and confirms the previously hypothesized distinction between affective and negative/disorganized dimensions of vulnerability to psychosis (*van Os and Kapur, 2009*; *van Os et al., 2010*). It is worth noting, however, that compared to our approach, factorial analysis separates symptoms that are considered to be the expression of distinct underlying latent variables. Therefore, by design, factorial analysis sacrifices information residing in the structure of correlations observed within and a cross large-scale dimensions (*Borsboom and Cramer, 2013*). By comparison, spatial embedding of individual symptoms captures the relationship between large-scale symptoms, such as the relative proximity of negative and disorganized dimensions, as well as the potential existence of relevant sub-clusters within large-scale dimensions. For instance, in both samples, avolition was located closer to affective and depressive symptoms compared to symptoms of reduced emotional expressiveness, which is in agreement with evidence of the existence of sub-dimensions within negative symptoms (*Kaiser et al., 2017*).

Aside from the structure of cross-sectional psychopathology, the key advantage of the MTSN approach is the ability to capture pathways of longitudinal interactions between symptoms. Indeed, despite the inherent dynamic nature of the 'Network Theory of Psychopathology' most network analyses are conducted on cross-sectional data, hence lacking the essential dimension of time. In our approach, the time dimension was intuitively captured in the second network component plotted along the vertical axis, with symptoms at baseline located at the bottom of the graph and symptoms at follow-up located at the top. Euclidean distance between symptoms offers therefore an intuitive characterization of the propensity of different clinical manifestations to influence one another across longitudinal assessments. For instance, according to the first cross-sectional dimension, negative symptoms of reduced emotional expressiveness were located in proximity to symptoms of conceptual disorganization and thought disturbances. However, in both samples, the second time dimension clearly distinguished between the two forms of psychopathology, with baseline symptoms of thought disturbance located much closer to psychopathology at follow-up compared to reduced emotional expressiveness. This finding would suggest that symptoms of reduced emotional expressiveness develop as a consequence of prior thought disturbance and disorganization, and have hence a less active role in influencing subsequent psychopathology. Such interpretation is consistent with the literature on basic symptoms of psychosis that suggests that subclinical subjectively experienced thought disturbances lie at the core of the phenomenology of the disorder and play an active role in influencing clinical evolution and particularly negative symptoms (*Schultze-Lutter et al., 2014*).

One of the main challenges in developmental and early intervention psychiatry is the growing realization that early clinical manifestations of psychopathology are largely not specific to a single clinical outcome. We propose that cross-diagnostic clinical evolutions may be related to specific mechanisms that act as developmental crossroads in the evolution of psychopathology. In particular, some clinical manifestations may broadly increase the risk for subsequent psychopathology, while others may broadly affect different forms of prior psychopathology. Targeting such symptoms where the 'flow' of psychopathology either broadens or narrows could be particularly effective in preventing deleterious clinical outcomes. The MLSN is ideally suited to identify such *gateways* and *funnels* of psychopathology, offering an intuitive characterization of longitudinal clinical pathways over time. For instance, in both samples, our analysis confirmed that sub-threshold manifestation of thought disturbance, acted as gateways, broadly increasing the risk for subsequent psychopathology. On the opposite, negative symptoms such as blunted affect and occupational functioning in 22q11DS or avolition-apathy in the NEURAPRO sample acted as funnels that were broadly passively influenced

by prior psychopathology. Moreover, some symptoms appeared to act as crossroads bridging across the affective to negative-disorganized dimensions over time. For instance, Hostility in the NEU-RAPRO sample was associated with thought disturbances at baseline but increased the risk for developing affective symptoms at follow-up. On the opposite, guilt in 22q11DS sample was associated with affective symptoms at baseline but increased the risk for both affective symptoms at thought disturbances at follow-up. Finally, particularly in the 22q11DS sample, our results pointed to an important role of reduced tolerance stress at follow-up, in firstly mediating the effects of prior effective disturbances on subsequent psychotic symptoms. These findings are strongly reminiscent of the reduced tolerance to stress in the 'affective pathway' to psychosis initially proposed by Myin-Germeys and Van Os (*Myin-Germeys et al., 2003*). Moreover, our findings also suggest that reduced tolerance to stress may partially mediate the effects of prior thought disorder on the subsequent development of affective disturbances. The prominent role of these pathways in 22q11DS may be related to recent evidence of dysregulation of the Hypothalamus-Pituitary-Adrenal-Axis (*Sandini et al., 2020*) and heightened vulnerability to environmental stress in this population (*Armando et al., 2018*).

Altogether results both in 22q11DS and NEURAPRO cohorts highlight the potentialities of an approach based on multilayer temporal network analysis to provide an intuitive and quantitative characterization of clinical pathways contributing to heterogenous clinical evolutions in the early stages of psychopathology.

## Predicting clinical evolution of individual patients through multilayer graph diffusion

Aside from shedding light on underlying disease mechanisms, a major appeal of understanding pathways of interaction between symptoms is in assisting in establishing prognosis. Still, current network approaches to psychopathology characterize symptoms exclusively in terms of their reciprocal connectivity profile, sacrificing information regarding symptom severity in individual participants. The unique feature of GSP is that network nodes are characterized not only in terms of connectivity, but can also be assigned a value or signal. For instance, in our GSP approach baseline symptoms were assigned a signal that corresponded to their observed severity in a particular individual. For each individual, we then predicted the evolution of psychopathology by modeling the diffusion of symptom severity from baseline to follow-up symptoms, as function of the structure of the multilayer temporal symptom network (see *Figure 2*).

To the best of our knowledge, our results are the first to demonstrate the potentialities of a purely network-based graph diffusion approach in predicting multivariate patterns of clinical evolution at the level of individual participants. Importantly, in both samples, prediction accuracy was significantly higher than simply considering clinical stability across time in both 22q11DS and NEURAPRO samples. It has been argued that an excessive focus on a single dichotomous clinical outcome such as conversion to psychosis might represent a major limit of the current UHR framework (*van Os and Guloksuz, 2017*). Indeed, the presence of a UHR status increases the likelihood of developing a range of psychopathological outcomes that have the potential to negatively influence an individual's functional outcome (*McGorry and van Os, 2013*; *McGorry et al., 2018*; *McGorry and Nelson, 2016*; *van Os and Guloksuz, 2017*). The potential for diverse psychiatric outcomes is also well described in individuals carrying genetic risk for psychosis (*Dean et al., 2010*) including in 22q11DS (*International Consortium on Brain and Behavior in 22q11.2 Deletion Syndrome et al., 2014*). Indeed, besides a 30% risk of developing a psychotic disorder individuals with 22q11DS present a 30% likelihood of presenting an anxiety disorder, a 30% likelihood of being diagnosed with ADHD, and a 20% risk of developing a mood disorder by adulthood, all of which can negatively affect the quality of life (*International Consortium on Brain and Behavior in 22q11.2 Deletion Syndrome et al., 2014*). A significant advantage of a network-based graph diffusion approach is that clinical prediction is performed at the level of individual symptoms with the potential of describing mixed and heterogeneous clinical evolutions.

Aside from flexibility in terms of considering clinical outcomes, the network-based graph diffusion approach is also flexible in terms of integrating multiple predictors at baseline. Indeed, results in both samples suggest that a broad clinical characterization at baseline, that goes beyond considering associations between homologous forms of psychopathology, can improve prediction of clinical outcome at follow-up. Specifically, in the NEURAPRO sample, prediction accuracy of CAARMS items

was improved when considering a combination of CAARMS and BPRS at baseline, whereas in the 22q11DS sample prediction, accuracy was strongly improved when adding a combination of BPRS and CBCL at baseline. Interestingly, network dimensionality reduction offered an intuitive appreciation of reasons underlying the value of additional clinical instruments in improving prediction accuracy. Indeed, the synergism of CBCL and BPRS was related to the fact that two instruments appeared to capture opposite facets of psychopathology, with CBCL assisting prediction of affective and ADHD symptoms while most BPRS items clustered closer to negative and disorganized aspects of the SIPS.

According to the model proposed by Van Os et al., these findings could imply that synergism between CBCL and BPRS is related to the fact that two instruments aid in prediction of two independent 'affective' and 'negative/disorganized' clinical pathways of vulnerability to psychosis (*Myin-Germeys and van Os, 2007*).

## Limitations and future directions

The present study should be considered as an exploratory attempt to address some of the significant challenges that are hindering the translation of network techniques approaches to the clinical setting. As such, results of this study should be considered in light of multiple significant limitations which remain to be addressed in future work.

A first significant limitation of the current manuscript is that several methodological differences across the two samples hinder the ability to directly compare results of network analysis across 22q11DS and non-syndromic clinical high-risk individuals. Indeed, different clinical instruments, different length of longitudinal follow, different therapeutic strategies, and different mean age across the two samples could all contribute to the observed difference in network structure. In this perspective, the interest of using independent cohorts was mostly to evaluate the potentialities of our methodological approach in a population that was less genetically and clinically homogenous than 22q11DS, more so than to directly compare candidate clinical pathways across samples.

From a methodological perspective, a significant limitation is that we did not explicitly test for the causal nature of the longitudinal interactions between symptoms. Hence, while the structure of such longitudinal correlations remains interesting from the clinical perspective of prognosis, conclusions regarding the existence of causal disease pathways between symptoms remain speculative. Multiple methodological techniques are currently being proposed to re-construct causal relationships between symptoms and have mostly been applied to clinical data measured with high-temporal resolution, using the Experience Sampling Method (ESM) (*Jordan et al., 2020*). Such techniques could potentially be fruitfully employed to test for causality between longitudinal clinical variables, measured at much lower temporal resolution, such as those analyses in the present study.

A further significant limitation is that, for both populations, we reconstructed a single symptom network in the entire sample. Therefore, the interpretations that can be drawn regarding the existence of different clinical pathways between individual symptoms are not personalized, but should rather be considered as referred to the entire population. However, weaker correlations between symptoms observed in the NEURAPRO sample compared to the 22q11DS sample, would suggest that additional factors could influence heterogeneous network structure in subgroups of individuals. The issue of how to allow network analysis techniques to capture sufficiently individualized information to actually inform clinical practice remains a significant issue, that is by no means fully addressed by the present study.

In our view, at least two current lines of research are particularly promising in terms of increasing the personalization of network analysis techniques. The first direction entails using network analysis to analyze clinical data collected with high temporal resolution using the ESM (*Borsboom and Cramer, 2013*; *Myin-Germeys et al., 2018*). Applying an MLSN approach to ESM data seems particularly promising as it could allow to capture individualized information regarding dynamic relations between symptoms as they occur in daily life (*Myin-Germeys et al., 2018*). However, the ESM approach is inherently limited in terms of its ability to capture causal relationships that occur across longer time-frames (i.e., months/years), that might be particularly clinically relevant in terms of guiding clinical decisions. In this perspective, a complementary approach to increase personalization in network analysis would consist in identifying subject-level moderators that influence the relationship between other variables in a network (*Haslbeck et al., 2021*). It might be particularly interesting to adapt such moderator analysis to a MultiLayer-Temporal-Symptom-Network, in order to attempt to

identify factors that influence the predictive value of specific longitudinal clinical pathways. For instance, it could be hypothesized that longitudinal clinical pathways might differ with age or across sexes. Testing such hypotheses in a network framework should in our view be the object of future work.

An additional significant limitation is that, while methods proposed in the present manuscript might contribute to increasing interpretability of psychiatric network analyses, results presented in the present manuscript could still prove excessively complex for clinical translation. In this perspective, we believe that network complexity is significantly and inherently related to the choice of number of nodes and edges that are represented, which remains however a largely arbitrary step in network analyses techniques (*Borsboom and Cramer, 2013*). Such arbitrariness arguably constitutes a significant limitation of current network analyses, compared to traditional consensus-based diagnostic approaches.

Finally, a significant limitation is that despite both cohorts being extensively phenotyped from the neurocognitive and neurobiological perspective, analyses presented in the present study were restricted to psycho-pathological variables. However, a wealth of literature indicates that clinical evolution in the early stages of psychosis is tightly related to subtle deviations in neurocognitive and neurodevelopmental trajectories (*Insel, 2010*), that are likely to influence pathways of interaction between symptoms detected by our analyses. Integrating such different dimensions remains a significant goal that should be addressed by future research.

An interesting future perspective is that network approaches are potentially extremely flexible for integrating data originating from different modalities, including for instance neuroimaging or genetics. Embedding a candidate biomarker in the context of longitudinal symptom network could offer an intuitive characterization of clinical variables that are affected. Moreover, the graph diffusion approach could allow to explicitly test the additive value of candidate biomarkers in terms of predictive clinical evolution. Indeed the benchmark against which any future biomarker should be tested is that of improving prediction achieved from gold standard clinical characterization instead of testing prediction performance independently from clinical scores (*Paulus, 2015*). Crucially, providing additive predictive values implies capturing processes that are not accessible to clinical evaluation more so than describing cross-sectional biomarkers that are strongly correlated with clinical scores, which has been the focus of most current genetic and neuroimaging research (*Kapur et al., 2012*). A pragmatic approach could be to investigate whether underlying neurobiological mechanisms are associated with differences in the structure of longitudinal symptoms network and hence improve the accuracy of graph diffusion-based prediction.

## Additional information

### Funding

| Funder | Grant reference number | Author |
| --- | --- | --- |
| Stanley Medical Research Institute | 07TGF-1102 | Barnaby Nelson<br>Paul G Amminger<br>Hok Pan Yuen<br>Connie Markulev<br>Monica R Schäffer<br>Nilufar Mossaheb<br>Monika Schlögelhofer<br>Stefan Smesny<br>Ian B Hickie<br>Gregor Emanuel Berger<br>Eric YH Chen<br>Lieuwe de Haan<br>Dorien H Nieman<br>Merete Nordentoft<br>Anita Riecher-Rössler<br>Swapna Verma<br>Andrew Thompson<br>Alison Ruth Yung<br>Patrick D McGorry |
| National Health and Medical Research Council | 566529 | Ian B Hickie<br>Alison Ruth Yung |

| | | Patrick D McGorry |
| --- | --- | --- |
| National Health and Medical Research Council | 1060996 | Patrick D McGorry |
| National Health and Medical Research Council | 1080963 | Alison Ruth Yung Patrick D McGorry |
| National Health and Medical Research Council | 566593 | Alison Ruth Yung Patrick D McGorry |
| National Health and Medical Research Council | 1027532 | Barnaby Nelson |
| Schweizerischer Nationalfonds zur Förderung der Wissenschaftlichen Forschung | FNS 320030_179404 | Stephan Eliez |
| Schweizerischer Nationalfonds zur Förderung der Wissenschaftlichen Forschung | FNS 324730_144260 | Stephan Eliez |
| Schweizerischer Nationalfonds zur Förderung der Wissenschaftlichen Forschung | PZ00P1_174206 | Maude Schneider |
| Schweizerischer Nationalfonds zur Förderung der Wissenschaftlichen Forschung | 51NF40-158776 | Stephan Eliez |

The funders had no role in study design, data collection and interpretation, or the decision to submit the work for publication.

## Author contributions

Corrado Sandini, Conceptualization, Formal analysis, Investigation, Methodology, Writing - original draft, Writing - review and editing; Daniela Zöller, Conceptualization, Formal analysis, Methodology, Writing - review and editing; Maude Schneider, Conceptualization, Data curation, Supervision, Writing - original draft, Writing - review and editing; Anjali Tarun, Formal analysis, Methodology, Writing - original draft, Writing - review and editing; Marco Armando, Conceptualization, Writing - review and editing; Barnaby Nelson, Conceptualization, Data curation, Supervision, Project administration, Writing - review and editing; Paul G Amminger, Data curation, Project administration, Writing - review and editing; Hok Pan Yuen, Data curation, Methodology, Project administration; Connie Markulev, Monica R Schäffer, Nilufar Mossaheb, Monika Schlögelhofer, Stefan Smesny, Ian B Hickie, Gregor Emanuel Berger, Eric YH Chen, Lieuwe de Haan, Dorien H Nieman, Merete Nordentoft, Anita Riecher-Rössler, Swapna Verma, Andrew Thompson, Alison Ruth Yung, Patrick D McGorry, Data curation, Project administration; Dimitri Van De Ville, Conceptualization, Formal analysis, Supervision, Validation, Methodology, Writing - review and editing; Stephan Eliez, Conceptualization, Supervision, Writing - review and editing

## Author ORCIDs

Corrado Sandini ⬚ https://orcid.org/0000-0003-2933-1607
Daniela Zöller ⬚ http://orcid.org/0000-0002-7049-0696
Maude Schneider ⬚ http://orcid.org/0000-0001-7147-8915
Dimitri Van De Ville ⬚ http://orcid.org/0000-0002-2879-3861

## Ethics

Clinical trial registration RCT; 'NEURAPRO'; trial registration: anzctr.org.au, identifier: 12608000475347.
Human subjects: Informed consent was obtained from all participants. In the 22q11DS sample the study was approved from the Ethics Board of the Geneva University. In the NEURAPRO sample the study was approved by the Melbourne Health Human Research Ethics Committee (HREC NO:2008.628).

Decision letter and Author response
Decision letter https://doi.org/10.7554/eLife.59811.sa1
Author response https://doi.org/10.7554/eLife.59811.sa2

## Additional files

### Supplementary files

• Supplementary file 1. Comparison of severity of CAARMS items at baseline between subjects excluded from the NEURAPRO cohort due to missing data and the rest of the sample.

• Supplementary file 2. Comparison of severity of CAARMS items at longitudinal follow-up between subjects excluded from the NEURAPRO cohort due to missing data and the rest of the sample.

• Source data 1. Source Data for Longitudinal Symptom Network in 22q11DS Sample.

• Source data 2. Source Data for Longitudinal Symptom Network in NEURAPRO Sample.

• Transparent reporting form

### Data availability

The data analyzed is clinically sensitive human data, derived from from an ongoing longitudinal study of children and adolescents affected from a neurogenic disorder and from a multi-centre double-blind clinical trial in psychiatric populations. Due to confidentiality agreements the data can not be publicly disseminated but is available, in anonymized format, upon direct reasonable request to Stephan Eliez, for the 22q11DS dataset, and Barnaby Nelson for the Neurapro Dataset.

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

# Appendix 1

## Supplementary analysis 1

We conducted supplementary analyses to verify that subjects who were excluded from the original NEURAPRO sample due to lack of data at either baseline or follow-up were clinically not significantly different from the rest of the cohort. Specifically, we compared excluded subjects (N=103) to the rest of the sample in terms of severity of available CAARMS items at both baseline and follow-up assessments using two-sample t-tests. P-value of differences was corrected for multiple comparisons using false discovery rate (FDR) correction at p<0.05 using Benjamini-Yekutieli procedure as implemented in Matlab. Results are resumed in *Supplementary file 1* for comparison of symptom severity at baseline. Comparison of symptom severity and longitudinal follow-up between included and excluded subjects are detailed in *Supplementary file 2*. In *Supplementary files 1* and *2*, we also report how many data points were available for each CAARMS item at baseline and follow-up assessment among subjects who were excluded. Results revealed that excluded subjects were not significantly different from the rest of the sample in terms of severity of any item the available CAARMS assessment at baseline or at longitudinal follow-up.

## Supplementary analysis 2
### Analysis of symptom predictability according to structure of multilayer symptom network

We followed the procedure employed in the work of Eiko Fried et al. (*Borsboom and Cramer, 2013*), using the available R Code (*Carpenter et al., 1985*), to compute the predictability of symptoms in our multilayer symptom network. Predictability was estimated for both symptoms at baseline and at follow-up considering both cross-sectional and longitudinal relationships between symptoms. In order to be able to compare predictability estimates across samples, we restricted our analysis to 38 symptoms that were available in both cohorts (15 items that were shared across SIPS and CAARMS and 23 BPRS items). To provide a visual representation of the differential predictability of different clinical variables, plotted predictability values as color-coding of symptoms in the multilayer network for both 22q11DS and NEURAPRO cohorts (see *Appendix 1—figure 1A and B*).

Results revealed that in both samples, negative symptoms had overall higher predictability both at baseline and at follow-up compared to the rest of symptoms in the network, contributing to an overall positive correlation between predictability values across samples (R-Pearson=0.22, p=0.054; see *Appendix 1—figure 1D*).

This analysis also revealed some differences in predictability patterns across samples. In particular, while in the 22q11DS sample, symptoms of disorganization and thought disturbance had high predictability both at baseline and at follow-up assessment in the NEURAPRO disorganization and thought disturbance symptoms had weaker predictability at baseline and higher predictability at follow-up assessment. Moreover, a comparison of predictability values across samples revealed an overall stronger predictability in the 22q11DS network compared to the NEURAPRO network (mean predictability in 22q11DS=0.64±0.26, mean predictability in NEURAPRO=0.44±0.20, p-value of paired t-test<0.0001; see *Appendix 1—figure 1C*).

As we highlight in the Limitation section of the study, interpretation of differences observed across samples is complicated by the presence of multiple potential confounding factors, such as different clinical instruments, different length of longitudinal follow, different therapeutic strategies, and different mean age across the two samples. A potential interpretation is that lower symptom predictability reflects higher clinical heterogeneity in the NEURAPRO sample compared to the 22q11DS participants, who share a common genetic predisposition to psychosis.

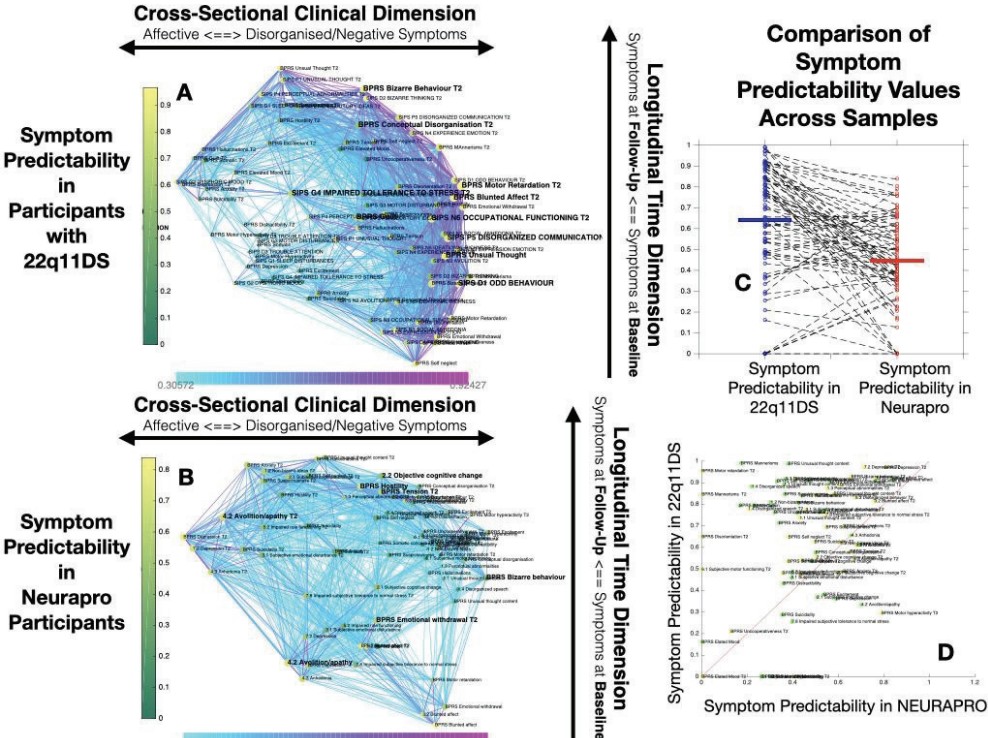

**Appendix 1—figure 1.** Comparison of Symptom Predictaiblity in 22q11DS and NEURAPRO Samples. (**A**) Representation of symptom predictability according to the multilayer temporal symptom network in the 22q11DS sample. Symptoms are spatially embedded according to the two main network dimensions derived from network eigendecompositon as described in the main text. As described in the main text, node size is scaled according to symptom connectivity strength. The specificity of this analysis is that node color is shaded according to network predictability, both symptoms at baseline and at follow-up. (**B**) Representation of symptom predictability according to the multilayer temporal symptom network in the 22q11DS sample. (**C**) Comparison of symptom predictability according to the structure of the multilayer symptom network across 22q11DS and NEURAPRO samples. Each point represents a predictability estimate for one symptom. Dashed lines connect predictability estimates for homologous symptoms across samples. (**D**) Comparison of symptom predictability according to the structure of the multilayer symptom network across 22q11DS and NEURAPRO samples. Each point represents a predictability estimate for one symptom. Symptoms at baseline are represented in green whereas symptoms at follow-up are represented in yellow. The red line separates symptoms with higher predictability in the 22q11DS sample from symptoms with higher predictability in the NEURAPRO sample.

## Supplementary analysis 3

In order to verify the impact of the ordinal nature on the structure of the symptoms network, we constructed multilayer symptom networks using Spearman rank correlations, for both the 22q11DS and NEURAPRO cohorts. First, we assessed whether the main dimensions derived from eigendecomposition of Spearmen correlations networks significantly differed from those of Pearson correlation networks. Specifically, we correlated loadings of symptoms according to the two main dimensions of Spearman versus Pearson correlation networks, for both the NEURAPRO and 22q11DS cohorts. In the NEURAPRO sample, we observed a very strong correlation between loadings of symptoms for both the first (R=0.98, p<0.001) and second network dimensions (R=0.97, p<0.001), indicating that the two main dimensions derived from Spearman versus Pearson correlation networks were almost identical.

Results in the 22q11DS cohort also point to a strong similarity in the main dimensions Spearman versus Pearson correlation networks as indicated by strong positive correlations between loadings of

symptoms according to the first (R=0.78, p<0.001) and second (R=0.85, p<0.001) network dimensions.

Next, we assessed whether longitudinal pathways of interaction between symptoms significantly differed in Spearman versus Pearson correlation networks by correlating measures of longitudinal betweenness centrality across the two types of networks for both samples. In the NEURAPRO sample, we observed a strong positive correlation between values of longitudinal betweenness centrality of symptoms in Spearman versus Pearson correlation networks (R=0.81, p<0.001). Moreover, the same symptoms were identified as the strongest longitudinal Hubs (Avolition, Bizarre Behavior, and Objective Cognitive Change) in both networks, indicating longitudinal clinical pathways in the two networks. We observed some differences across the two networks in symptoms that were identified as weaker longitudinal hubs. Specifically, BPRS symptoms of Hostility at baseline and Emotional Withdrawal at follow-up were identified as hubs only in the Pearson network whereas centrality of BPRS Uncooperativeness was significant only in the Spearman network, after FDR correction for multiple comparisons.

In the 22q11DS sample, we also observed a strong positive correlation between values of longitudinal betweenness centrality of symptoms in Spearman versus Pearson correlation networks (R=0.83, p<0.001), with most symptoms being identified as longitudinal hubs in both networks. Similarly, to the NEURAPRO sample, we observed some discrepancies in terms of which of the weaker longitudinal hubs was considered significant after correction for multiple comparisons. Specifically, conceptual disorganization at follow-up was significant only in the Pearson correlation network. Baseline symptoms of suspiciousness and impaired tolerance and follow-up symptoms of somatic disturbance and mannerisms were significant only in Spearman networks.

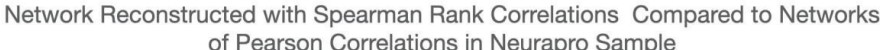

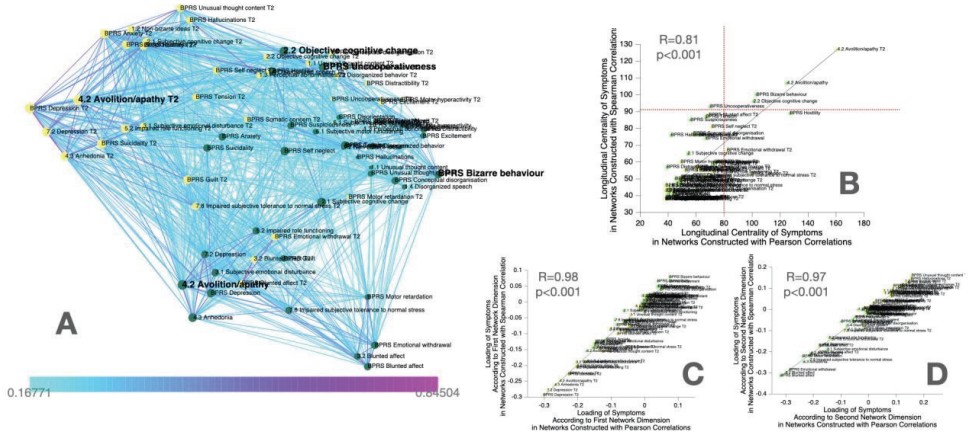

**Appendix 1—figure 2.** Construction and comparison of multilayer symptom network constructed using Spearman's rank correlation compared to Pearson correlation in the NEURAPRO sample. (**A**) Multilayer symptom network reconstructed from Spearman rank correlations in the NEURAPRO sample, employing the same procedure as described in the main text. (**B**) Correlation of longitudinal betweenness centrality measures computed in networks constructed using Spearman versus Pearson correlations. The horizontal red dashed line separates symptoms that had higher longitudinal centrality estimates than expected by chance, in the Spearman correlation networks. The vertical dashed line separates symptoms that had higher longitudinal centrality estimates than expected by chance, in the Pearson correlation network. (**C**) Correlation of loading of symptoms across the first dimension of Spearman versus Pearson correlation networks in the NEURAPRO sample. (**D**) Correlation of loading of symptoms across the second dimension of Spearman versus Pearson correlation networks in the NEURAPRO sample.

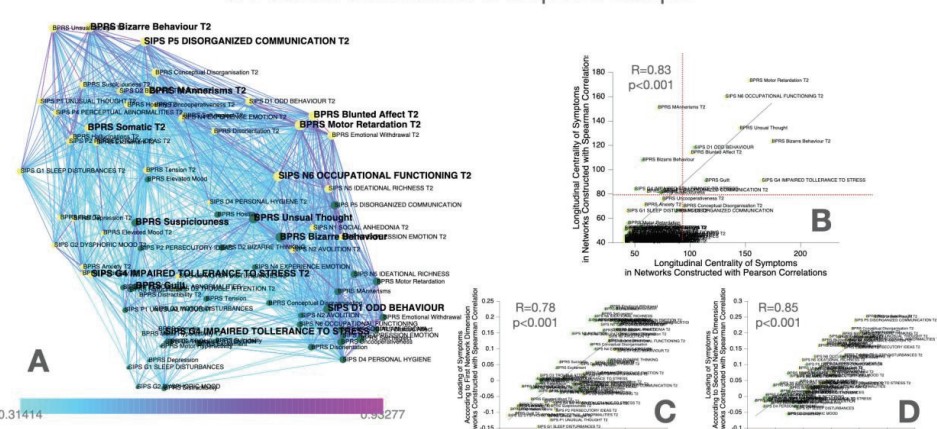

**Appendix 1—figure 3.** Construction and comparison of multilayer symptom network constructed using Spearman's rank correlation compared to Pearson correlation in the 22q11DS sample. (**A**) Multilayer symptom network reconstructed from Spearman rank correlations in 22q11DS, employing the same procedure as described in the main text. (**B**) Correlation of longitudinal betweenness centrality measures computed in networks constructed using Spearman versus Pearson correlations. The horizontal red dashed line separates symptoms that had higher longitudinal centrality estimates than expected by chance, in the Spearman ccorrelation networks. The vertical dashed line separates symptoms that had higher longitudinal centrality estimates than expected by chance, in the Pearson correlation network. (**C**) Correlation of loading of symptoms across the first dimension of Spearman versus Pearson correlation networks in the 22q11DS sample. (**D**) Correlation of loading of symptoms across the second dimension of Spearman versus Pearson correlation networks in the 22q11DS sample.

## Supplementary analysis 4
### Methods

We conducted complementary analyses to assess the stability of results, after pruning networks to include only the most significant connections, over a range of connectivity strength thresholds.

To select the appropriate range of thresholds, we employed the following procedure. For the most lenient threshold, we considered all correlations between symptoms that were significant at p<0.05 after FDR correction of multiple comparisons. Subsequently, we progressively removed 2% of all possible connections in the network, starting from the least statistically significant ones, until we reached a threshold where the network was no longer fully connected. In the 22q11DS cohort, this procedure yielded a range of six thresholds going from 41% to 31% of all possible connections in the network, while in the NEURAPRO cohort, the procedure yielded a range of five thresholds going from 32% to 24% of all possible connections.

We next repeated the procedure for multilayer temporal network analysis described in the manuscript, including network dimensionality reduction and graph theory analysis, for each of these thresholds. Resulting networks for each threshold are represented in *Appendix 1—figures 4* and *5*.

In order to assess the stability of network dimensionality reduction, we computed Pearson correlations of symptom loading across the first and the second network dimension for each couple of network thresholds (i.e., correlations of symptoms loadings for networks reconstructed considering 32% vs. 26% of network edges). For five possible thresholds, these 10 individual correlation values, which quantified the stability of network dimensionality reduction across thresholds. To have an estimate of the overall stability across multiple thresholds, we computed the mean and the standard deviation of such Pearson correlation values.

Moreover, for each network threshold, we computed longitudinal betweenness centrality values for each symptom as described in the Materials and methods section of the study. Longitudinal betweenness centrality values were then correlated across multiple network thresholds to quantify their stability as described above.

## Results

In the 22q11DS cohort, networks were constructed for six different thresholds (41%, 39%, 37%, 35%, 33%, and 31% of all possible connections). The mean Pearson correlation coefficient of symptom loadings across five thresholds was 0.996±0.0016 for the first network dimension and 0.996±0.002 for the second network dimension. The mean Pearson correlation coefficient for longitudinal betweenness centrality was 0.988±0.0061.

In the NEURAPRO cohort, networks were constructed for five different thresholds (32%, 30%, 28%, 26%, and 24% of all possible connections). The mean Pearson correlation coefficient of symptom loadings across five thresholds was 0.994±0.0039 for the first network dimension and 0.996±0.002 for the second network dimension. The mean Pearson correlation coefficient for longitudinal betweenness centrality was 0.991±0.0046.

## Conclusion

These results indicated that in both cohorts, network dimensionality and graph theory analysis of longitudinal clinical pathways were highly stable across a range of network connection thresholds. We report such results in the supplementary section of the study.

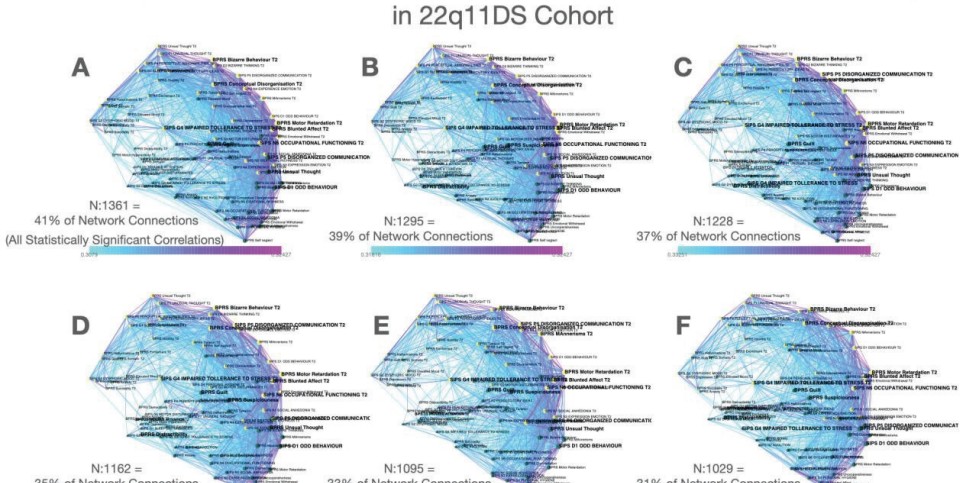

**Appendix 1—figure 4.** Stability of networks constructed considering multiple thresholds of connectivity strength in the 22q11DS cohort. (A–F) Multilayer temporal symptom networks constructed considering a range of connectivity strength thresholds. For the most lenient threshold, we considered all correlations that were significant at p<0.05 after FDR correction for multiple comparisons (Panel A, 41% of possible connections). We progressively pruned 2% of the least significant connections until reaching the most stringent connectivity threshold that still yielded a fully connected network (Panel F, 31% of possible connections).

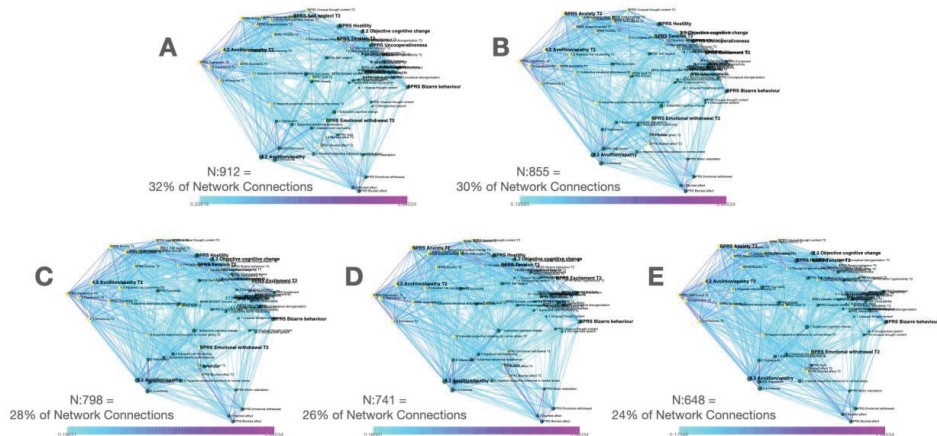

Appendix 1—figure 5. Stability of networks constructed considering multiple thresholds of connectivity strength in the NEURAPRO cohort. (A–D) Multilayer temporal symptom networks constructed considering a range of connectivity strength thresholds. For the most lenient threshold, we considered all correlations that were significant at p<0.05 after FDR correction for multiple comparisons(Panel A, 32% of possible connections). We progressively pruned 2% of the least significant connections until reaching the most stringent connectivity threshold that still yielded a fully connected network (Panel E, 24% of possible connections).

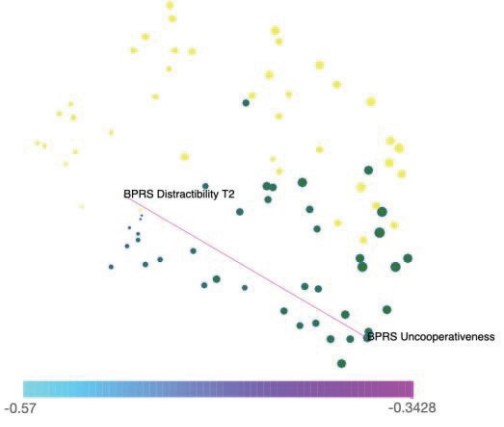

Appendix 1—figure 6. Representation of only significant negative correlations between symptoms in the 22q11DS sample. Network visualization of significant negative correlations between symptoms in the 22q11DS cohort. Spatial embedding of symptoms is performed according to the eigendecomposition of networks composed of both significant positive and negative correlations. Results in 22q11DS indicated a single negative correlation that was significant at p<0.05 after FDR correction for multiple comparisons, connecting BPRS assessment of uncooperativeness at baseline assessment with BRPS assessment of distractibility and longitudinal follow-up.

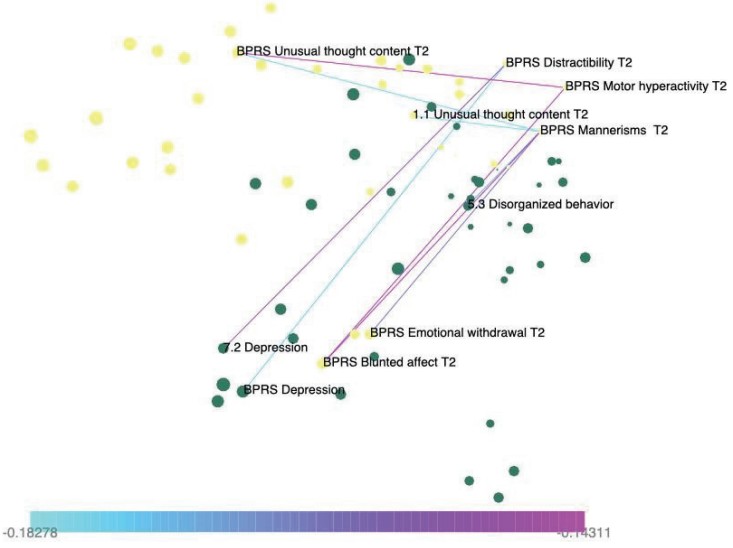

**Appendix 1—figure 7.** Representation of only significant negative correlations between symptoms in the NEURAPRO sample. Network visualization of significant negative correlations between symptoms in the NEURAPRO cohort. Spatial embedding of symptoms is performed according to the eigendecomposition of networks composed of both significant positive and negative correlations. Results in NEURAPRO indicated a total of eight negative correlations that were significant at p<0.05 after FDR correction for multiple comparisons. Such negative correlations mainly connected symptoms of blunted affect with symptoms of distractibility and motor hyperactivity.

## Procedure to select the appropriate number of dimensions after network eigendecomposition

In order to choose the appropriate number of network dimensions, we examined the percentage of variance explained by each component derived from network eigendecomposition. Percentage of explained variance of each network component is displayed in *Appendix 1—figure 8B* for the 22q11DS sample and *Appendix 1—figure 9B* for the NEURAPRO sample. In the 22q11DS cohort, the first component explained 32.7% of the variance in the network, whereas the second, third, and fourth components explained 30.3%, 7.1%, and 4.5% of the variance, respectively. In the NEURAPRO cohort, the first four components explained 23.8%, 14.4%, 8.5%, and 6.26% of the network variance.

In order to have a principled criterion to choose the number of meaningful components, we repeated dimensionality reduction after randomly reshuffling the position of network edges, for 1000 iterations. This procedure yielded a distribution of variance explained by the first dimension of a network composed of the same connection, but with random structure. This analysis revealed that the first component of a randomized network explained between 5.9% and 4.8% of network structure in the 22q11DS sample and between 6.5% and 4.5% of network structure in the NEURAPRO sample. Minimum and maximum percentage of variance explained by the main dimension of a

randomized network are displayed as dashed red lines, in *Appendix 1—figure 8B* for the 22q11DS sample and *Appendix 1—figure 9B* for the NEURAPRO sample. We considered network components derived from eigendecomposition of empirical networks to be significant only if they explained more variance than the strongest component of a random network.

This analysis revealed that in both 22q11DS and NEURAPRO samples, only the first three network dimensions explained a higher amount of variance of the first dimension of a randomized network. However, while the first two dimensions explained more than twice the amount of variance compared to what observed in a randomized network, the third dimension explained only marginally higher variance compared to what would be expected in a random network.

In order to verify whether the third network dimension still captured potentially clinically significant features of network structure, we embed symptoms considering the first and third network dimensions in both populations. A representation of network embedding according to the first and third dimensions can be found in *Appendix 1—figure 8A* for the 22q11DS sample and *Appendix 1—figure 9A* for the NEURAPRO sample. Finally, we correlated empirically observed correlation strength between symptoms with their distance according to the third network dimension, separately for cross-sectional and longitudinal correlation, as described in the main text. Results of this final analysis are displayed in *Appendix 1—figure 8C* for the 22q11DS sample and *Appendix 1—figure 9C* for the NEURAPRO sample.

These analyses revealed that in the NEURAPRO sample, the third network dimension mainly separated baseline symptoms from follow-up symptoms across time. However, this third dimension did not accurately capture variance in correlation strength among different longitudinal correlations as indicated by a non-significant correlation of Euclidean distance and correlation strength (R=0.03, p=0.1).

Similarly, in the 22q11DS sample, the third network dimension tended to separate baseline from follow-up symptoms but did not capture variance between longitudinal correlations as indicated by a non-significant correlation between Euclidean distance across this third dimension and correlation strength (R=0.03, p=0.1).

This means that, for both samples, symptoms at baseline and follow-up that were closer to one another across this third network dimension were not actually more strongly correlated. These results imply that the third dimension did not offer a meaningful representation of network structure. Taken together, these results justify the use of the first two network dimensions for topological embedding of symptoms, as reported in the main text.

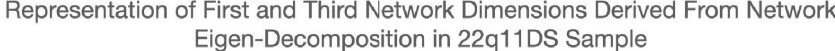

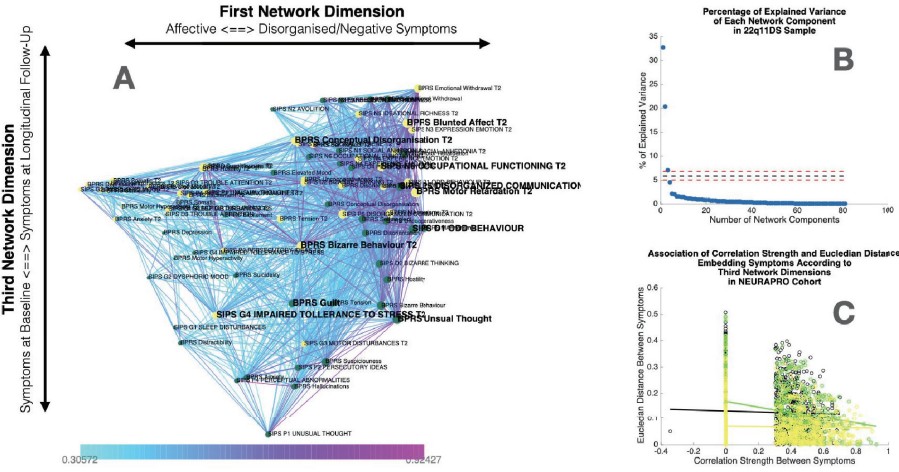

**Appendix 1—figure 8.** Representation of first and third network dimensions derived from network eigendecomposition in the 22q11DS sample. (**A**) Network spatial embedding according to the first network dimension along the horizontal axis and third network dimension along the vertical axis in

*Appendix 1—figure 8 continued on next page*

*Appendix 1—figure 8 continued*

the 22q11DS sample. The third network dimension mainly separated symptoms at baseline assessment, displayed in green and mostly found at the bottom of the graph, from symptoms at follow-up assessment displayed in yellow and mostly found on the top of the graph. (**B**) Representation of percentage of network variance explained by each network component from the first components on the left to the last components on the right. Red dashed lines indicate the minimum and maximum percentage of variance explained by eigendecomposition of a network composed of 1000 networks composed of the same edges but with randomized structure. Network components located below the dashed lines are considered to be non-meaningful as they explain less percentage of network structure than what observed in a network with random structure. (**C**) Correlation of Euclidean distance between symptoms according to the third network dimension and empirically observed correlation strength between symptoms. The association of Euclidean distance and correlation strength is computed separately for cross-sectional correlations between symptoms at baseline assessment displayed in green and between symptoms at longitudinal follow-up displayed in yellow, as well as for longitudinal correlations between symptoms across time displayed in black.

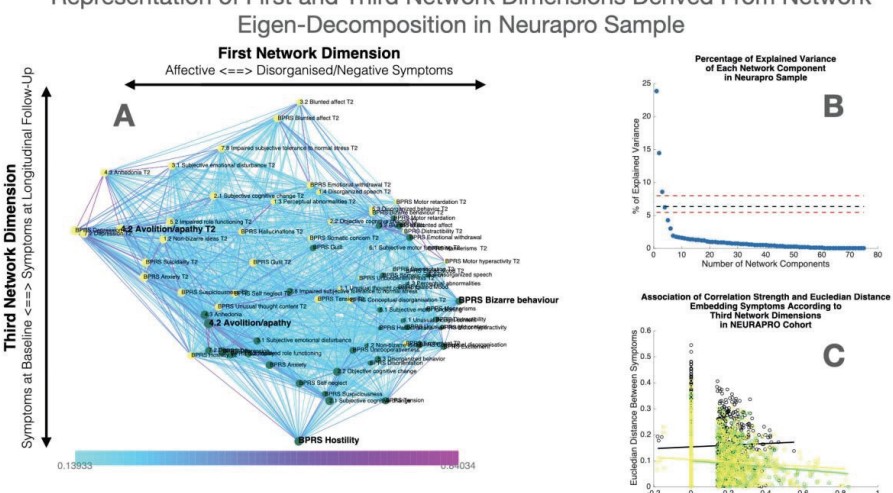

**Appendix 1—figure 9.** Representation of first and third network dimensions derived from network eigendecomposition in the NEURAPRO sample. (**A**) Network spatial embedding according to the first network dimension along the horizontal axis and third network dimension along the vertical axis in NEURAPRO sample. The third network dimension mainly separated symptoms at baseline assessment, displayed in green and mostly found at the bottom of the graph, from symptoms at follow-up assessment displayed in yellow and mostly found on the top of the graph. (**B**) Representation of percentage of network variance explained by each network component from the first components on the left to the last components on the right. Red dashed lines indicate the minimum and maximum percentage of variance explained by eigendecomposition of a network composed of 1000 networks composed of the same edges but with randomized structure. Network components located below the dashed lines are considered to be non-meaningful as they explain less percentage of network structure than what observed in a network with random structure. (**C**) Correlation of Euclidean distance between symptoms according to the third network dimension and empirically observed correlation strength between symptoms. The association of Euclidean distance and correlation strength is computed separately for cross-sectional correlations between symptoms at baseline displayed in green and at longitudinal follow-up displayed in yellow, as well as for longitudinal correlations between symptoms across time displayed in black.

## Negative correlations

In order to verify that a wide data range was not significantly contributing to driving the correlation between Euclidean distance and correlation strength, we repeated this analysis after restricting the data range by excluding negative correlations. This analysis confirmed the presence of a significant negative correlation between Euclidean distance and correlation strength. Indeed, significant correlations were observed for both cross-sectional correlations between symptoms at baseline (R=−0.354, p<0.001 in 22q11DS, R=−0.247, p<0.001 in NEURAPRO), for cross-sectional correlations between symptoms at longitudinal follow-up (R=−0.473, p<0.001 in 22q11DS, R=0.279, p<0.001 in NEURAPRO), and for longitudinal correlations connections symptoms across time (R=−0.365, p<0.001 in 22q11DS, R=−0.154, p<0.001 in NEURAPRO).

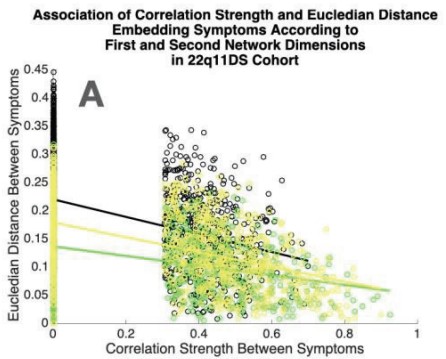
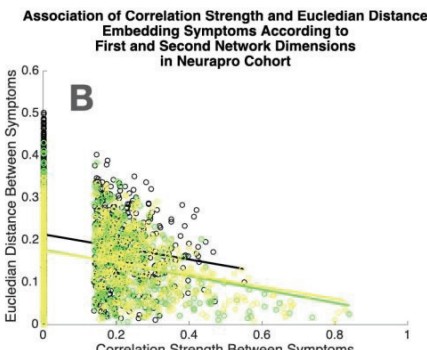

**Appendix 1—figure 10.** Association of Euclidean distance and correlation strength not considering negative correlations. (**A**) Association of Euclidean distance between symptoms according to the first two network dimensions and empirically observed correlation strength, not considering negative correlations, in the 22q11DS sample. (**B**) Association of Euclidean distance between symptoms according to the first two network dimensions and empirically observed correlation strength, not considering negative correlations, in the NEURAPRO sample. For both samples, the association of Euclidean distance and correlation strength is computed separately for cross-sectional correlations between symptoms at baseline displayed in green and at longitudinal follow-up displayed in yellow, as well as for longitudinal correlations between symptoms across time displayed in black.

