## [Decision Letter]

**Acceptance summary:**

There is a growing interest in using network science approaches to understand psychopathology. Yet the existing body of evidence has some important limitations. Prior work has focused on cross-sectional data, and has focused on group effects. Moreover, existing approaches lack the intuitiveness of more traditional descriptions. Leveraging 2 high-risk psychosis samples, the authors propose new approaches aimed at overcoming these important limitations, with the goal of providing the emergence of illness at the level of the individual, a key step toward making precision medicine a clinical reality.

**Decision letter after peer review:**

Thank you for submitting your article "Characterization and Prediction of Clinical Pathways of Vulnerability to Psychosis through Graph Signal Processing" for consideration by *eLife*. Your article has been reviewed by 2 peer reviewers, and the evaluation has been overseen by a Reviewing Editor and Richard Ivry as the Senior Editor. The reviewers have opted to remain anonymous.

Summary

There is a growing interest in using network science approaches to understand psychopathology. Yet the existing body of evidence has some important limitations. Prior work has focused on cross-sectional data, and has focused on group effects. Moreover, existing approaches lack the intuitiveness of more traditional descriptions. Leveraging 2 high-risk psychosis samples, the authors propose new approaches aimed at overcoming these important limitations, with the goal of providing the emergence of illness at the level of the individual, a key step toward making precision medicine a clinical reality.

The reviewers and I identified several strengths to your submission:

• The data are presented in a clear and easy-to-understand fashion and the paper is, principally, well-argued and clearly worthy of publication.

• I appreciate the discussion of network replicability (cf. Forbes et al. J Abnormal Psychol, 2017)

• I appreciate the focus on emergence and complex (dynamic) system. This is important. It has been discussed for many years (see Meehl and Sellars, 1956) and has long played a role in psychological science (e.g., cognitive science – consider the PDP workers, Rumelhart and McClelland; imagery workers, e.g., Kosslyn). In psychopathology, Depue and Lenzenweger (2001). The focus on emergent properties in psychological phenomena has a long and rich history predating McNally and Co. as well as others interested in network analysis.

• The paper employs methods from network science to investigate psychopathology. Strengths of the paper include the use of temporal network methods, dimensionality reduction, as well as signal diffusion across graphs, all of which are applied to rich clinical data. The results are interesting and of potential interest.

Nevertheless, a number of substantive concerns significantly dampened enthusiasm

Essential revisions:

• Introduction – There is insufficient motivation for the use and significance of network models. Both reviewers highlighted this as a major limitation of the manuscript. Network modeling has become somewhat fashionable in psychopathology research owing to its newness and quantitative allure. The authors, however, do not provide a compelling rationale for using this technique. Can the authors show a substantial increment or special leverage offered by this method? What theoretical or scientific question is at stake here? What gain is provided by this approach beyond a different way of explaining the empirical relationships amongst the variables analyzed?

• Method. As written, it was challenging to discern the rationale for the specific methods and, ultimately, the significance of the findings. Both reviewers highlighted this as a major limitation. Network analysis is a relatively new set of techniques that many readers may not be familiar with. And, even for those familiar with these techniques, a lot of critical information is left out and/or described only in the supplementary materials. Beyond that, many of the analyses lack motivation and the metrics provided lack adequate description (e.g., What does strength represent and why is it important?). Because of these issues, in its current form, it will be hard for readers to glean what the findings mean and why they're important. The stability of the solution is also a concern.

[Editors' note: further revisions were suggested prior to acceptance, as described below.]

Thank you for resubmitting your work entitled "Characterization and Prediction of Clinical Pathways of Vulnerability to Psychosis through Graph Signal Processing" for further consideration by *eLife*. Your revised article has been reviewed by 2 peer reviewers and the evaluation has been overseen by Drs. Ivry as the Senior Editor and Shackman as the Reviewing Editor.

The manuscript has been improved but there is one significant issue that needs to be addressed.

We would like you to provide a sober and slightly more comprehensive discussion of the limitations of the study, and some concrete suggestions for future research aimed at addressing those limitations. To quote Reviewer 3, "I believe it would be valuable to extend the section on study limitations to state that the investigation reflects an initial attempt at tacking the difficult questions addressed in a relatively exploratory fashion."

---

## [Author Response]

Essential revisions:• Introduction – There is insufficient motivation for the use and significance of network models. Both reviewers highlighted this as a major limitation of the manuscript. Network modeling has become somewhat fashionable in psychopathology research owing to its newness and quantitative allure. The authors, however, do not provide a compelling rationale for using this technique. Can the authors show a substantial increment or special leverage offered by this method? What theoretical or scientific question is at stake here? What gain is provided by this approach beyond a different way of explaining the empirical relationships amongst the variables analyzed?

We thank the reviewers for this pertinent commentary. We have significantly re-structured the introduction to address their concerns.

Specifically, before introducing network approaches to psychopathology we have introduced a section of the introduction where we more explicitly delineate the limitation of current dimensional and categorical approaches, firstly in terms of their underlying conceptualization of mental health disturbances and secondly from the perspective of guiding clinical practice.

“The first approach based on diagnostic categories is intuitive, and has proven extremely useful in increasing communicability and agreement across clinicians [1]. […] Indeed, it is increasingly recognized that, in psychiatry, symptoms are not only passive expression of common underlying disease processes, but can often represent active agents, that have the ability to provoke their reciprocal emergence, through dynamic causal interactions [5, 6]. For instance….”

We have added a section where we detail the overall objective of developing novel data analysis techniques for psychiatry, namely embracing the complexity of clinical manifestations and allowing for a personalized approach to clinical care.

“Clinical evidence presented above points to two considerations, which will be relevant for designing novel approaches to the assessment and classification of patients. […] The hope is that tools of data-science will allow to embrace the complexity of mental health problems, and ultimately assist in a personalized approach to clinical care [10].”

We have also provided a more thorough description of the potentialities of network approaches to the study of psychopathology, with regards to the limitation that were previously highlighted for current dimensional and categorical approaches.

Specifically, we have added a section to describe the core computational appeal of network analysis approaches: namely the ability to characterize broad elements of the overall structure of a data-set while conserving highly granular information relative to individual variables.

“Network science is a rapidly expanding branch of mathematics dedicated to the study of graphs, which can be broadly defined as structures composed of discrete nodes that are connected by edges [12]. […] For instance, the propensity for and individual variable to strongly mediate the relationship between other network variables, can defined as a network centrality.”

Moreover, we have added a section to describe how computational advantages of network analysis could translate to an improved understanding of mental health disturbances.

“The application of network analysis carries the potential have a profound theoretical and practical impact on the study of mental health disturbances. […] High centrality is commonly considered to reflect of a prominent causal role in influencing other symptoms.”

Finally, we have added a section to provide concrete examples of insights that are currently generated through the application of network analysis techniques to the extended psychosis phenotype.

“Network approaches are rapidly gaining popularity by demonstrating that exploiting high-dimensional granularity of clinical assessments can generate insights that would be missed if symptoms were merged in diagnoses or dimensions [17]. […] Finally, by using network analysis, studies have shown that clinical manifestations that are not necessarily specific to psychosis, including in particular affective dysregulation, may play a prominent role in influencing the presence of psychotic-like experiences (PLE), in adolescence [21].”

• Method. As written, it was challenging to discern the rationale for the specific methods and, ultimately, the significance of the findings. Both reviewers highlighted this as a major limitation. Network analysis is a relatively new set of techniques that many readers may not be familiar with. And, even for those familiar with these techniques, a lot of critical information is left out and/or described only in the supplementary materials. Beyond that, many of the analyses lack motivation and the metrics provided lack adequate description (e.g., What does strength represent and why is it important?). Because of these issues, in its current form, it will be hard for readers to glean what the findings mean and why they're important. The stability of the solution is also a concern.

We thank the reviewers for this pertinent commentary. Indeed, we agree that the rational for several procedures described in the methods section was not sufficiently elaborated, while some methodical details are not sufficiently detailed. We have significantly restructured several sections of the methods to address these concerns.

Specifically, the sections detailing the procedure for network topological embedding was adapted to integrate and introductory section detailing the rational and objective for conducting network dimensionality reduction. We have also added a section to justify the choice of employing only the two main network dimensions for topological embedding of networks.

Moreover, we have added a section to describe the rational for computing correlations between Euclidian distance and correlation strength, and what is the underlying hypothesis.

“Network topological embedding

Arguably one of the main challenges of current network models relates to intuitiveness and interpretability of results. […] Moreover, we correlated topological embedding of symptoms according to the two main network dimensions across samples, in order to have an estimate of the degree of similarity of network structure across samples.”

The section describing graph theory analysis was also completely restructured. We have added an introductory section detailing the objectives of performing graph theoretical analysis in the first place. Moreover, we have improved the description of the specific graph theoretical measures that were employed, both in terms of the underlying rational and interpretation of each measure (i.e meaning of connectivity strength or betweenness centrality) and the specific algorithms that were employed to derive the measures. The section now reads as follows:

“Graph theory analysis of longitudinal clinical pathways

Spatial embedding of symptoms provided an intuitive representation of the major patterns of relationships between symptoms. […] Such longitudinal network hubs at follow-up can be conceptualized as *funnels* of psychopathology.”

[Editors' note: further revisions were suggested prior to acceptance, as described below.]

The manuscript has been improved but there is one significant issue that needs to be addressed.We would like you to provide a sober and slightly more comprehensive discussion of the limitations of the study, and some concrete suggestions for future research aimed at addressing those limitations. To quote Reviewer 3, "I believe it would be valuable to extend the section on study limitations to state that the investigation reflects an initial attempt at tacking the difficult questions addressed in a relatively exploratory fashion."

We thank the reviewers for this pertinent commentary. Indeed, as it was previously formulated, we agree that the manuscript did not sufficiently portray the extent of the limitations of our work, nor it’s exploratory nature. We have significantly reformulated and expanded the limitations section in an effort to address this concern. Specifically, compared to the previous version, we have highlighted additional limitations of our approach. Moreover, we chose to integrate the limitations and future directions section, in an effort to provide a more coherent discussion of future work that will be necessary in order address the limitations of the present study.

Firstly, we highlight the fact that we did not explicitly test for causality in the clinical pathways identified by our analysis as a significant limitation of our work. Compared to the previous version we also add a reference describing recent methodological developments that have been proposed to reconstruct causality in interactions between psychiatric symptoms. Such techniques have mostly been applied to data acquired with high-temporal resolution, using the Experience Sampling Method (ESM). We suggest that future work could attempt to apply such techniques to estimate causality in clinical pathways occurring with much lower temporal dynamics, such as those described in the present study.

“From a methodological perspective, a significant limitation is that we did not explicitly test for the causal nature of the longitudinal interactions between symptoms. […] Such techniques could potentially be fruitfully employed to test for causality between longitudinal clinical variables, measured at much lower temporal resolution, such as those analyses in the present study.”

Compared to the previous version, we highlight the lack of network personalization as a significant limitation of work. Specifically, we explicitly state that reconstructing a single network for the entire sample implies that interpretations regarding the existence of different clinical pathways, should be referred to the entire sample, and are not therefore personalized. We highlight the fact that the poverty of subject-level information remains a main limit to current network analysis techniques, that is by no means fully addressed by the present study.

**“**A further significant limitation is that, for both populations, we reconstructed a single symptom network in the entire sample. […] The issue of how to allow network analysis techniques to capture sufficiently individualized information to actually inform clinical practice remains a significant issue, that is by no means fully addressed by the present study.”

Compared to the previous version of the manuscript we have also added a section detailing our views regarding two promising research directions that could be proposed to address the insufficient personalization of current network analysis techniques. We suggest ways in which work presented in the present study could potentially inform such efforts.

Specifically, we suggest that a Multilayer-Symptom-Network approach could be employed to analyze data collected using the Experience Sampling Method (ESM), which could allow to depict personalized information regarding pathways of interaction between different clinical variables as they occur in daily life. Moreover, we suggest that introduction of Moderated Network Analysis, could be a promising perspective to identify subject level predictors of specificities in the structure of longitudinal symptom networks.

“In our view, at least two current lines of research are particularly promising in terms of increasing personalization of network analysis techniques. […] Testing such hypotheses in a network framework should in our view be the object of future work.”

Compared to the previous version of the manuscript, we have further explicitly stated two additional limitations which we highlighted during the review process. Such limitations refer to the observation that, despite our efforts, MLSN presented in the study might still prove excessively complex for clinical translation. Moreover, we suggest that arbitrariness in the choice of number of nodes and edges is a further unresolved issue in current network analyses, that is both tightly related to the complexity of resulting networks, and a significant limitation compared to traditional consensus-based diagnostic approaches.

**“**An additional significant limitation is that, while methods proposed in the present manuscript might contribute to increasing interpretability of psychiatric network analyses, results presented in the present manuscript could still prove excessively complex for clinical translation. […] Such arbitrariness arguably constitutes a significant limitation of current network analyses, compared to traditional consensus-based diagnostic approaches.”

Finally, we highlight the fact that our analyses were restricted to psychopathological variables, which represents in our view a significant limitation of our work. Indeed, we believe that underlying neurobiological processes, that are potentially captured by the extensive phenotyping performed in both 22q11DS and NEURAPRO cohorts, could play a significant role in influencing clinical pathways of interactions between symptoms.

In this perspective we suggest some approaches through which the MLSN approach could be employed to integrate different dimensions, such as cognition, genetics and neurobiology, that are involved in the clinical evolution of individuals in the early stages of psychopathology.

**“**Finally, a significant limitation is that despite both cohorts being extensively phenotyped from the neurocognitive and neurobiological perspective, analyses presented in the present study were restricted to psycho-pathological variables. […] A pragmatic approach could be to investigate whether underlying neurobiological mechanisms are associated with differences in the structure of longitudinal symptoms network and hence improve accuracy of graph-diffusion based prediction.”